# Recursive Reasoning in Minimax Games: A Level $k$ Gradient Play Method

**Zichu Liu**
University of Toronto
jieben.liu@mail.utoronto.ca

**Lacra Pavel**
University of Toronto
pavel@ece.utoronto.ca

## Abstract

Despite the success of generative adversarial networks (GANs) in generating visually appealing images, they are notoriously challenging to train. In order to stabilize the learning dynamics in minimax games, we propose a novel recursive reasoning algorithm: Level $k$ Gradient Play (Lv.$k$ GP) algorithm. In contrast to many existing algorithms, our algorithm does not require sophisticated heuristics or curvature information. We show that as $k$ increases, Lv.$k$ GP converges asymptotically towards an accurate estimation of players' future strategy. Moreover, we justify that Lv.$\infty$ GP naturally generalizes a line of provably convergent game dynamics which rely on predictive updates. Furthermore, we provide its local convergence property in nonconvex-nonconcave zero-sum games and global convergence in bilinear and quadratic games. By combining Lv.$k$ GP with Adam optimizer, our algorithm shows a clear advantage in terms of performance and computational overhead compared to other methods. Using a single Nvidia RTX3090 GPU and 30 times fewer parameters than BigGAN on CIFAR-10, we achieve an FID of 10.17 for unconditional image generation within 30 hours, allowing GAN training on common computational resources to reach state-of-the-art performance.

## 1   Introduction

In recent years, there has been a surge of powerful models that require simultaneous optimization of several objectives. This increasing interest in multi-objective optimization arises in various domains - such as generative adversarial networks [21, 27], adversarial attacks and robust optimization [37, 7], and multi-agent reinforcement learning [36, 54] - where several agents aim at minimizing their objectives simultaneously. Games generalize this optimization framework by introducing different objectives for different learning agents, known as players. The generative adversarial network is a widely-used method of this type, which has demonstrated state-of-the-art performance in a variety of applications, including image generation [28, 51], image super-resolution [32], and image-to-image translation [10]. Despite their success at generating visually appealing images, GANs are notoriously challenging to train [40, 39]. Naive application of the gradient-based algorithm in GANs often leads to poor image diversity (sometimes manifesting as "mode collapse") [40], Poincare recurrence [39], and subtle dependency on hyperparameters [18]. An immense corpus of work is devoted to exploring and enhancing the stability of GANs, including techniques as diverse as the use of optimal transport distance [1, 22], critic gradient penalties [53], different neural network architectures [26, 6], feature matching [49], pre-trained feature space [50], and minibatch discrimination [49]. Nevertheless, architectural modifications (e.g., StyleGANs [27]) require extensive computational resources, and many theoretically appealing methods (Follow-the-ridge [55], CGD[52]) require Hessian inverse operations, which is infeasible for most GAN applications.

To stabilize the learning dynamics in GANs, many recent efforts rely on sophisticated heuristics that allow the agents to anticipate each other's next move [16, 52, 23]. This anticipation is an example of a recursive reasoning procedure in cognitive science [12]. Similar to how humans think, recursive

36th Conference on Neural Information Processing Systems (NeurIPS 2022).

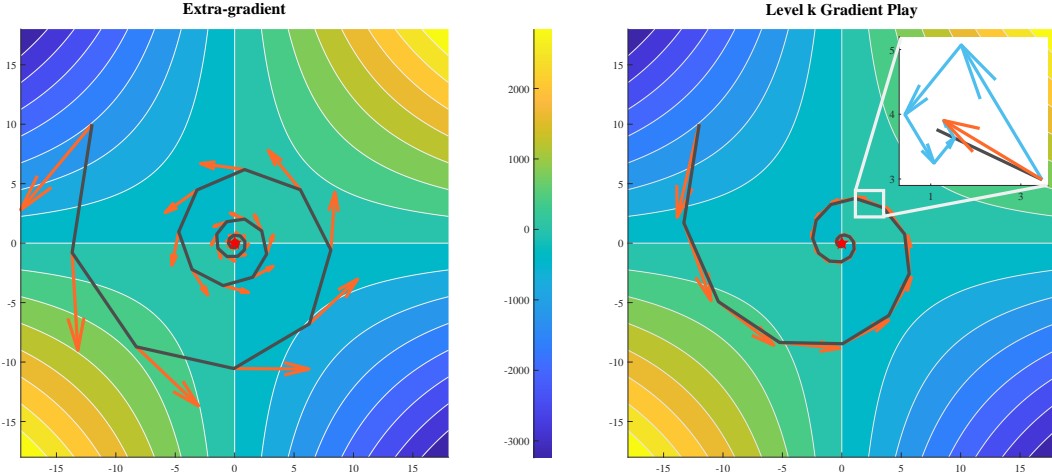

Figure 1: Illustration of predictive algorithms on: $\min_x \max_y 10xy$. Left: Extra-gradient algorithm. Right: Level $k$ gradient play algorithm ($k = 6$). The solution, trajectory $\{x_t, y_t\}_{t=1}^T$ and anticipated future state are shown with red star, black line and orange arrow, resp. The subplot in the right figure depicts how Lv.$k$ GP predicts future states by showing its reasoning procedure with blue arrows. More steps in the reasoning process leading to better anticipations and faster convergence.

reasoning represents the belief reasoning process where each agent considers the reasoning process of other agents, based on which it expects to make better decisions. Importantly, it enables the use of opponents that reason about the learning agent, rather than assuming fixed opponents; the process can therefore be nested in a form as 'I believe that you believe that I believe...'. Based on this intuition, we introduce a novel recursive reasoning algorithm that utilizes only gradient information to optimize GANs. Our contributions can be summarized as follows:

(i) We propose a novel algorithm: level $k$ gradient play (Lv.$k$ GP), which is capable of reasoning about players' future strategy. In a game, agents at Lv.$k$ adjust their strategies in accordance with the strategies of Lv.$k - 1$ agents. We justify that, while typical GANs optimizers, such as Learning with Opponent Learning Awareness (LOLA) and Symplectic Gradient Adjustment (SGA), approximate Lv.2 and Lv.3 GP, our algorithm permits higher levels of strategic reasoning. In addition, the proposed algorithm is amenable to neural network optimizers like Adam [29].

(ii) We show that, in smooth games, Lv.$k$ GP converges asymptotically towards an accurate prediction of agents' next move. Under mutual opponent shaping, two Lv.$\infty$ agents will naturally have a consistent view of one another if the Lv.$k$ GP converges as $k$ increases. Based on this idea, we provide a closed-form solution for Lv.$\infty$ GP: the Semi-Proximal Point Method (SPPM).

(iii) We prove the local convergence property of Lv.$\infty$ GP in nonconvex - nonconcave zero-sum games and its global convergence in bilinear and quadratic games. The theoretical analysis we present indicates that strong interactions between competing agents can increase the convergence rate of Lv.$k$ GP agents in a zero-sum game.

(iv) By combining Lv.$k$ GP with Adam optimizer, our algorithm shows a clear advantage in terms of performance and computational overhead compared to other methods. Using a single 3090 GPU with 30 times fewer parameters and 16 times smaller mini-batches than BigGAN [6] on CIFAR-10, we achieve an FID score [24] of 10.17 for unconditional image generation within 30 hours, allowing GAN training on common computing resources to reach state-of-the-art performance.

## 2    Related Works

In recent years, minimax problems have attracted considerable interest in machine learning in light of their connection with GANs. Gradient descent ascent (GDA), a generalization of gradient descent for minimax games, is the principal approach for training GANs in applications. GDA alternates between a gradient descent step for the min-player and a gradient ascent step for the max-player. The

convergence of GDA in games is far from as well understood as gradient descent in single-objective problems. Despite the impressive image quality generated by GANs, GDA fails to converge even in bilinear zero-sum games. Recent research on GDA has established a unified picture of its behavior in bilinear games in continuous and discrete-time [39, 46, 13, 41, 20]. First, [39] revealed that continuous-time GDA dynamics in zero-sum games result in Poincare recurrence, where agents return arbitrarily close to their initial state infinitely many times. Second, [3, 57] examined the discrete-time GDA dynamics, showing that simultaneous update of two players results in divergence while the agents' strategies remain bounded and cycle when agents take turns to update their strategies.

The majority of existing approaches to stabilizing GDA follow one of three lines of research. The essence of the first method is that the discriminator is trained until convergence while the generator parameters are frozen. As long as the generator changes slowly enough, the discriminator still converges in the presence of small generator perturbations. The two-timescale update rule proposed by [21, 24, 42] aims to keep the discriminator's optimality while updating the generator at an appropriate step size. [25, 14] proved that this two-timescale GDA with finite timescale separation converges towards the strict local minimax/Stackelberg equilibrium in differentiable games. [15, 55] explicitly find the local minimax equilibrium in games with secon-order optimization algorithms.

The second line of research overcomes the failure of GDA in games with predictive updates. Extra-gradient method (EG) [30] and optimistic gradient descent (OGD) [13] use the predictability of the agents' strategy to achieve better convergence property. Their variants are developed to improve the training performance of GANs [8, 19, 43]. [45] provided a unified analysis of EG and OGD, showing that they approximate the classical proximal point methods. Competitive gradient descent [52] models the agents' next move by solving a regularized bilinear approximation of the underlying game. Learning with opponent learning awareness (LOLA) and consistent opponent learning awareness (COLA) [16, 56] introduced opponent shaping to this problem by explicitly modeling the learning strategy of other agents in the game. LOLA models opponents as naive learners rather than LOLA agents, while COLA utilizes neural networks to predict opponents' next move. Lookahead-minimax [9] stabilizes GAN training by 'looking ahead' at the sequence of future states generated by an inner optimizer. In the game theory literature, recent work has proposed Clairvoyant Multiplicative Weights Update (CMWU) for regret minimization in general games [47]. Although CMWU is proposed to solve finite normal form games, which are different from unconstrained continuous games that Lv.$k$ GP aims to solve, both CMWU and Lv.$k$ GP share the same motivation of enabling the learning agent to update their strategy based on the opponent's future strategy. From this aspect, Lv.$k$ GP can be viewed as a specialized variant of CMWU that is specific to the problem of two-player zero-sum games, but adapted for unconstrained continuous kernel games.

Other methods directly modify the GDA algorithm with ad-hoc modifications of game dynamics and introduction of additional regularizers. Consensus optimization (CO) [40] and gradient penalty [22, 53] improve convergence by directly minimizing the magnitude of players' gradients. Symplectic gradient adjustment (SGA) [33, 4, 17] improves convergence by disentangling convergent potential components from rotational Hamiltonian components of the vector field.

## 3 Preliminaries

### 3.1 Notation

In this paper, vectors are lower-case bold letters (e.g. $\boldsymbol{\theta}$), matrices are upper-case bold letters (e.g. $\boldsymbol{A}$). For a function $f : \mathbb{R}^d \to \mathbb{R}$, we denote its gradient by $\nabla f$. For functions of two vector arguments $f(\boldsymbol{x}, \boldsymbol{y}) : \mathbb{R}^{d_1} \times \mathbb{R}^{d_2} \to \mathbb{R}$ we use $\nabla_x f, \nabla_y f$ to denote its partial gradients. We use $\nabla_{xx} f, \nabla_{yy} f, \nabla_{xy} f$ to denote its Hessian. A stationary point of $f$ denotes the point where $\nabla_x f = \nabla_y f = \boldsymbol{0}$. We use $\|\boldsymbol{v}\|$ to denote the Euclidean norm of vector $\boldsymbol{v}$. We refer to the largest and smallest eigenvalues of a matrix $\boldsymbol{A}$ by $\lambda_{\max}(\boldsymbol{A})$ and $\lambda_{\min}(\boldsymbol{A})$, respectively. Moreover, we denote the spectral radius of matrix $\boldsymbol{A}$ by $\rho(\boldsymbol{A}) = \max\{|\lambda_1|, \ldots, |\lambda_n|\}$, i.e., the eigenvalue with largest absolute value.

### 3.2 Problem Definition

In order to justify the effectiveness of recursive reasoning procedure, in this paper, we consider the problem of training Generative Adversarial Networks (GANs)[21]. The GANs training strategy defines a two-player game between a generative neural network $G_{\boldsymbol{\theta}}(\cdot)$ and a discriminative neural network $D_{\boldsymbol{\phi}}(\cdot)$. The generator takes as input random noise $\boldsymbol{z}$ sampled from a known distribution

$\mathbf{P_z}$, e.g., $\mathbf{z} \sim \mathbf{P_z}$, and outputs a sample $G_{\boldsymbol{\theta}}(\mathbf{z})$. A discriminator takes as input a sample $\boldsymbol{x}$ (either sampled from the true distribution $\mathbf{P_x}$ or from the generator) and attempts to classify it as real or fake. The goal of the generator is to fool the discriminator. The optimization of GAN is formulated as a two-player differentiable game where the generator $G_{\boldsymbol{\theta}}$ with parameter $\boldsymbol{\theta}$, and the discriminator $D_{\boldsymbol{\phi}}$ with parameters $\boldsymbol{\phi}$, aim at minimizing their own cost function $f(\boldsymbol{\theta}, \boldsymbol{\phi})$ and $g(\boldsymbol{\theta}, \boldsymbol{\phi})$ respectively, as follows:

$$\min_{\boldsymbol{\theta} \in \mathbb{R}^m} f(\boldsymbol{\theta}, \boldsymbol{\phi}) \text{ and } \min_{\boldsymbol{\phi} \in \mathbb{R}^n} g(\boldsymbol{\theta}, \boldsymbol{\phi}), \tag{1}$$

where the two function $f$ and $g : \mathbb{R}^m \times \mathbb{R}^n \to \mathbb{R}$. When $f = -g$ the corresponding optimization problem is called a two-player zero-sum game and it becomes a minimax problem:

$$\min_{\boldsymbol{\theta} \in \mathbb{R}^m} \max_{\boldsymbol{\phi} \in \mathbb{R}^n} f(\boldsymbol{\theta}, \boldsymbol{\phi}). \tag{Minimax}$$

In this work, we assume the cost functions have Lipschitz continuous gradients with respect to all model parameters $(\boldsymbol{\theta}, \boldsymbol{\phi})$:

**Assumption 3.1.** *The gradient $\nabla_{\boldsymbol{\theta}} f(\boldsymbol{\theta}, \boldsymbol{\phi})$, is $L_{\boldsymbol{\theta\theta}}$−Lipschitz with respect to $\boldsymbol{\theta}$ and $L_{\boldsymbol{\theta\phi}}$−Lipschitz with respect to $\boldsymbol{\phi}$ and the gradient $\nabla_{\boldsymbol{\phi}} g(\boldsymbol{\theta}, \boldsymbol{\phi})$, is $L_{\boldsymbol{\phi\phi}}$−Lipschitz with respect to $\boldsymbol{\phi}$ and $L_{\boldsymbol{\phi\theta}}$−Lipschitz with respect to $\boldsymbol{\theta}$, i.e.,*

$$\|\nabla_{\boldsymbol{\theta}} f(\boldsymbol{\theta}_1, \boldsymbol{\phi}) - \nabla_{\boldsymbol{\theta}} f(\boldsymbol{\theta}_2, \boldsymbol{\phi})\| \leq L_{\boldsymbol{\theta\theta}} \|\boldsymbol{\theta}_1 - \boldsymbol{\theta}_2\| \text{ for all } \boldsymbol{\phi},$$
$$\|\nabla_{\boldsymbol{\theta}} f(\boldsymbol{\theta}, \boldsymbol{\phi}_1) - \nabla_{\boldsymbol{\theta}} f(\boldsymbol{\theta}, \boldsymbol{\phi}_2)\| \leq L_{\boldsymbol{\theta\phi}} \|\boldsymbol{\phi}_1 - \boldsymbol{\phi}_2\| \text{ for all } \boldsymbol{\theta},$$
$$\|\nabla_{\boldsymbol{\phi}} g(\boldsymbol{\theta}_1, \boldsymbol{\phi}) - \nabla_{\boldsymbol{\phi}} g(\boldsymbol{\theta}_2, \boldsymbol{\phi})\| \leq L_{\boldsymbol{\phi\theta}} \|\boldsymbol{\theta}_1 - \boldsymbol{\theta}_2\| \text{ for all } \boldsymbol{\phi},$$
$$\|\nabla_{\boldsymbol{\phi}} g(\boldsymbol{\theta}, \boldsymbol{\phi}_1) - \nabla_{\boldsymbol{\phi}} g(\boldsymbol{\theta}, \boldsymbol{\phi}_2)\| \leq L_{\boldsymbol{\phi\phi}} \|\boldsymbol{\phi}_1 - \boldsymbol{\phi}_2\| \text{ for all } \boldsymbol{\theta}.$$

*We define $L := \max\{L_{\boldsymbol{\theta\theta}}, L_{\boldsymbol{\theta\phi}}, L_{\boldsymbol{\phi\theta}}, L_{\boldsymbol{\phi\phi}}\}$.*

## 4 Level $k$ Gradient Play

In this section, we propose a novel recursive reasoning algorithm, Level $k$ Gradient Play (Lv.$k$ GP), that allows the agents to discover self-interested strategies while taking into account other agents' reasoning processes. In Lv.$k$ GP, $k$ steps of recursive reasoning is applied to obtain an anticipated future state $(\boldsymbol{\theta}_t^{(k)}, \boldsymbol{\phi}_t^{(k)})$, and the current states $(\boldsymbol{\theta}_t, \boldsymbol{\phi}_t)$ are then updated as follows:

$$\text{Reasoning: } \begin{cases} \boldsymbol{\theta}_t^{(n)} = \boldsymbol{\theta}_t - \eta \nabla_{\boldsymbol{\theta}} f(\boldsymbol{\theta}_t, \boldsymbol{\phi}_t^{(n-1)}) \\ \boldsymbol{\phi}_t^{(n)} = \boldsymbol{\phi}_t - \eta \nabla_{\boldsymbol{\phi}} g(\boldsymbol{\theta}_t^{(n-1)}, \boldsymbol{\phi}_t) \end{cases} \quad \text{Update: } \begin{cases} \boldsymbol{\theta}_{t+1} = \boldsymbol{\theta}_t^{(k)} \\ \boldsymbol{\phi}_{t+1} = \boldsymbol{\phi}_t^{(k)} \end{cases} \tag{Lv.k GP}$$

We define the current state $(\boldsymbol{\theta}_t, \boldsymbol{\phi}_t)$, to be the starting point $(\boldsymbol{\theta}_t^{(0)}, \boldsymbol{\phi}_t^{(0)})$, of the reasoning process. Learning agents that adopt Lv.$k$ GP strategy are then called Lv.$k$ agents. Lv.1 agents act naively in response to the current state using GDA dynamics and Lv.2 agents act in response to Lv.1 agents by assuming its opponent as a naive learner. Therefore, Lv.$k$ GP allows for higher levels of strategic reasoning. The inspiration comes from how humans collaborate: humans are great at anticipating how their actions will affect others, so they frequently find out how to collaborate with other people to reach a "win-win" solution. The key to human collaboration is their ability to understand how other humans think which helps them develop strategies that benefit their collaborators. One of our main theoretical results is the following theorem, which demonstrates that agents adopting Lv.$k$ GP can precisely predict other players' next move and reach a consensus on their future strategies:

**Theorem 4.1.** *Suppose Assumption 3.1 holds. Let us define $\boldsymbol{\omega}_t = [\boldsymbol{\theta}_t, \boldsymbol{\phi}_t]^T \in \mathbb{R}^{m+n}$, $\boldsymbol{\omega}_t^{(k)} = [\boldsymbol{\theta}_t^{(k)}, \boldsymbol{\phi}_t^{(k)}]^T \in \mathbb{R}^{m+n}$ and $\Delta_{\max} = 2 \times \max(\|\nabla_{\boldsymbol{\theta}} f(\boldsymbol{\theta}_t, \boldsymbol{\phi}_t)\|, \|\nabla_{\boldsymbol{\phi}} g(\boldsymbol{\theta}_t, \boldsymbol{\phi}_t)\|)$. Assume $\boldsymbol{\omega}_t^{(k)}$ lie in a complete subspace of $\mathbb{R}^{m+n}$. Then for Lv.k GP we have:*

$$\|\boldsymbol{\omega}_t^{(k)} - \boldsymbol{\omega}_t^{(k-1)}\| \leq \eta \cdot (\eta L)^{(k-1)} \Delta_{\max}, \tag{2}$$

*Suppose the learning rate satisfies: $\eta < (2L)^{-1}$, then the sequence $\{\boldsymbol{\omega}_t^{(k)}\}_{k=0}^{\infty}$ is a Cauchy sequence. That is, given $\epsilon > 0$, there exists $N$ such that, if $a > b > N$ then:*

$$\|\boldsymbol{\omega}_t^{(a)} - \boldsymbol{\omega}_t^{(b)}\| < \mathcal{O}(\eta^b) < \epsilon \tag{3}$$

*Moreover, the sequence $\{\boldsymbol{\omega}_t^{(k)}\}_{k=0}^{\infty}$ converges to a limit $\boldsymbol{\omega}_t^*$: $\lim_{k \to \infty} \boldsymbol{\omega}_t^{(k)} = \boldsymbol{\omega}_t^*$.*

In accordance with Theorem 4.1, if we define $\boldsymbol{\omega}_{t+1} = \boldsymbol{\omega}_t^*$, then Lv.$\infty$ GP is equivalent to the following implicit algorithm where we call it Semi-Proximal Point Method:

$$\begin{cases} \boldsymbol{\theta}_{t+1} = \boldsymbol{\theta}_t - \eta\nabla_{\boldsymbol{\theta}}f(\boldsymbol{\theta}_t, \boldsymbol{\phi}_{t+1}) \\ \boldsymbol{\phi}_{t+1} = \boldsymbol{\phi}_t - \eta\nabla_{\boldsymbol{\phi}}g(\boldsymbol{\theta}_{t+1}, \boldsymbol{\phi}_t) \end{cases} \tag{SPPM}$$

### 4.1 Algorithms as an Approximation of SPPM

SPPM players arrive at a consensus by knowing precisely what their opponents' future strategies will be. Existing algorithms are not able to offer this kind of agreement. For instance, consensus optimization[40] forces the learning agents to cooperate regardless of their own benefits. Agents employ extra-gradient method[30], SGA[4], and LOLA[16] consider their opponents as naive learners, ignoring their strategic reasoning ability. CGD[52] takes into account the reasoning process of learning agents; however, it leads to an inaccurate prediction of agents in games that have cost functions with non-zero higher order derivatives ($n \geq 3$) [56]. In this section, we consider a subset of provably convergent variants of GDA in the Minimax setting, showing that, for specific choice of hyperparameters, the mentioned algorithms either approximate SPPM or approximate the approximations of SPPM:

Table 1: The update rules for the first player of SGA, LOLA, Lv.2 GP, LEAD, CGD, Lv.3 GP and SPPM in a Minimax problem and their precision as an approximation of SPPM. [†]The usage of zero or negative momentum has been suggested in recent works [20, 18]. For sake of comparison, we assume no momentum factor in LEAD's update, which corresponds to $\beta = 0$ in Equation 10 of [23].

| Algorithm | Update Rule | Precision |
|---|---|---|
| SGA [4] | $\boldsymbol{\theta}_{t+1} = \boldsymbol{\theta}_t - \eta\nabla_{\boldsymbol{\theta}}f(\boldsymbol{\theta}_t, \boldsymbol{\phi}_t) - \eta\gamma\nabla_{\boldsymbol{\theta}\boldsymbol{\phi}}f(\boldsymbol{\theta}_t, \boldsymbol{\phi}_t)\nabla_{\boldsymbol{\phi}}f(\boldsymbol{\theta}_t, \boldsymbol{\phi}_t)$ | —— |
| LOLA [16] | $\boldsymbol{\theta}_{t+1} = \boldsymbol{\theta}_t - \eta\nabla_{\boldsymbol{\theta}}f(\boldsymbol{\theta}_t, \boldsymbol{\phi}_t) - \eta\delta\nabla_{\boldsymbol{\theta}\boldsymbol{\phi}}f(\boldsymbol{\theta}_t, \boldsymbol{\phi}_t)\nabla_{\boldsymbol{\phi}}f(\boldsymbol{\theta}_t, \boldsymbol{\phi}_t)$ | —— |
| Lv.2 GP | $\boldsymbol{\theta}_{t+1} = \boldsymbol{\theta}_t - \eta\nabla_{\boldsymbol{\theta}}f(\boldsymbol{\theta}_t, \boldsymbol{\phi}_t + \eta\nabla_{\boldsymbol{\phi}}f(\boldsymbol{\theta}_t, \boldsymbol{\phi}_t))$ | $\mathcal{O}(\eta^2)$ |
| LEAD[†] [23] | $\boldsymbol{\theta}_{t+1} = \boldsymbol{\theta}_t - \eta\nabla_{\boldsymbol{\theta}}f(\boldsymbol{\theta}_t, \boldsymbol{\phi}_t) - \alpha\nabla_{\boldsymbol{\theta}\boldsymbol{\phi}}f(\boldsymbol{\theta}_t, \boldsymbol{\phi}_t)(\boldsymbol{\phi}_t - \boldsymbol{\phi}_{t-1})$ | —— |
| CGD[1] [52] | $\boldsymbol{\theta}_{t+1} = \boldsymbol{\theta}_t - \eta\nabla_{\boldsymbol{\theta}}f(\boldsymbol{\theta}_t, \boldsymbol{\phi}_t) - \eta\nabla_{\boldsymbol{\theta}\boldsymbol{\phi}}f(\boldsymbol{\theta}_t, \boldsymbol{\phi}_t)(\boldsymbol{\phi}_{t+1} - \boldsymbol{\phi}_t)$ | $\mathcal{O}(\eta^3)$ |
| Lv.3 GP | $\boldsymbol{\theta}_{t+1} = \boldsymbol{\theta}_t - \eta\nabla_{\boldsymbol{\theta}}f(\boldsymbol{\theta}_t, \boldsymbol{\phi}_t + \eta\nabla_{\boldsymbol{\phi}}f(\boldsymbol{\theta}_t - \eta\nabla_{\boldsymbol{\theta}}f(\boldsymbol{\theta}_t, \boldsymbol{\phi}_t), \boldsymbol{\phi}_t))$ | $\mathcal{O}(\eta^3)$ |
| SPPM (Lv.$\infty$ GP) | $\boldsymbol{\theta}_{t+1} = \boldsymbol{\theta}_t - \eta\nabla_{\boldsymbol{\theta}}f(\boldsymbol{\theta}_t, \boldsymbol{\phi}_{t+1})$ | 0 |

In Table 1, we compare the orders of precision of different algorithms as an approximation of SPPM in Minimax games with infinitely differentiable objective functions. In accordance with Equation (3) of Theorem 4.1, Lv.k GP is an $\mathcal{O}(\eta^k)$ approximation of SPPM. In order to analyze how well existing algorithms approximate SPPM, we consider the first-order Taylor approximation to SPPM:

$$\begin{cases} \boldsymbol{\theta}_{t+1} = \underbrace{\boldsymbol{\theta}_t - \eta\nabla_{\boldsymbol{\theta}}f(\boldsymbol{\theta}_t, \boldsymbol{\phi}_t) - \eta^2\nabla_{\boldsymbol{\theta}\boldsymbol{\phi}}f(\boldsymbol{\theta}_t, \boldsymbol{\phi}_t)\nabla_{\boldsymbol{\phi}}f(\boldsymbol{\theta}_t, \boldsymbol{\phi}_t)} - \eta^2\nabla_{\boldsymbol{\theta}\boldsymbol{\phi}}f(\boldsymbol{\theta}_t, \boldsymbol{\phi}_t)\nabla_{\boldsymbol{\phi}\boldsymbol{\theta}}f(\boldsymbol{\theta}_t, \boldsymbol{\phi}_t)(\boldsymbol{\theta}_{t+1} - \boldsymbol{\theta}_t) \\ \boldsymbol{\phi}_{t+1} = \boldsymbol{\phi}_t + \eta\nabla_{\boldsymbol{\theta}}f(\boldsymbol{\theta}_t, \boldsymbol{\phi}_t) - \eta^2\nabla_{\boldsymbol{\phi}\boldsymbol{\theta}}f(\boldsymbol{\theta}_t, \boldsymbol{\phi}_t)\nabla_{\boldsymbol{\theta}}f(\boldsymbol{\theta}_t, \boldsymbol{\phi}_t) - \eta^2\nabla_{\boldsymbol{\phi}\boldsymbol{\theta}}f(\boldsymbol{\theta}_t, \boldsymbol{\phi}_t)\nabla_{\boldsymbol{\theta}\boldsymbol{\phi}}f(\boldsymbol{\theta}_t, \boldsymbol{\phi}_t)(\boldsymbol{\phi}_{t+1} - \boldsymbol{\phi}_t) \end{cases}$$

$$1^{st} \text{ order approximation of Lv.2 GP}$$

Under-brace terms correspond to the first-order Taylor approximation of Lv.2 GP. For an appropriate choice of hyperparameters, SGA ($\gamma = \eta$) and LOLA ($\delta = \eta$) are identical to the first-order Taylor approximation of Lv.2 GP, where each agent models their opponent as a naive learner. Hence, we list them above the Lv.2 GP, which approximates SPPM up to $\mathcal{O}(\eta^2)$. CGD exactly recovers the first-order Taylor approximation of SPPM. In games with cost functions that have non-negative higher order derivatives ($n \geq 3$), the remaining term in SPPM's first-order approximation is an error of magnitude $\mathcal{O}(\eta^3)$, which means that CGD's accuracy is in the same range as that of Lv.3 GP. In bilinear and quadratic games where the objective function is at most twice differentiable, CGD

---

[1] The CGD update for the max player $\boldsymbol{\phi}$ is $\boldsymbol{\phi}_{t+1} = \boldsymbol{\phi}_t + \eta\nabla_{\boldsymbol{\phi}}f(\boldsymbol{\theta}_t, \boldsymbol{\phi}_t) + \eta\nabla_{\boldsymbol{\phi}\boldsymbol{\theta}}f(\boldsymbol{\theta}_t, \boldsymbol{\phi}_t)(\boldsymbol{\theta}_{t+1} - \boldsymbol{\theta}_t)$. If we substitute $(\boldsymbol{\phi}_{t+1} - \boldsymbol{\phi}_t)$ into $\boldsymbol{\theta}$'s update (and substitute $(\boldsymbol{\theta}_{t+1} - \boldsymbol{\theta}_t)$ into $\boldsymbol{\phi}$'s update, respectively), we arrive at the first-order approximation of SPPM. See A.5 for derivation details.

is equivalent to SPPM. The distinction between LEAD ($\alpha = \eta$) and CGD can be understood by considering their update rules. LEAD is an explicit method where opponents' potential next strategies are anticipated based on their most recent move ($\phi_t - \phi_{t-1}$). On the contrary, CGD accounts for this anticipation in an implicit manner, ($\phi_{t+1} - \phi_t$), where the future states appear in current states' update rules. Therefore, the computation of CGD updates requires solving a function involving additional Hessian inverse operations. A numerical justification is also provided in Table 2, showing that the approximation accuracy of Lv.$k$ GP improves as $k$ increases.

## 5 Convergence Property

In Theorem 4.1, we have analytically proved that Lv.$k$ GP convergences asymptotically towards SPPM, we will use this result to study the convergence property of Lv.$k$ GP in games based on our analysis of SPPM. The local convergence of SPPM in a non convex - non concave game can be analyzed via the spectral radius of the game Jacobian around a stationary point:

**Theorem 5.1.** *Consider the (Minimax) problem under Assumption 3.1 and Lv.k GP. Let $(\boldsymbol{\theta}^*, \boldsymbol{\phi}^*)$ be a stationary point. Suppose $\boldsymbol{\theta}_t - \boldsymbol{\theta}^*$ not in kernel of $\nabla_{\phi\theta} f(\boldsymbol{\theta}^*, \boldsymbol{\phi}^*)$, $\boldsymbol{\phi}_t - \boldsymbol{\phi}^*$ not in kernel of $\nabla_{\theta\phi} f(\boldsymbol{\theta}^*, \boldsymbol{\phi}^*)$ and $\eta < (L)^{-1}$. There exists a neighborhood $\mathcal{U}$ of $(\boldsymbol{\theta}^*, \boldsymbol{\phi}^*)$ such that if SPPM started at $(\boldsymbol{\theta}_0, \boldsymbol{\phi}_0) \in \mathcal{U}$, the iterates $\{\boldsymbol{\theta}_t, \boldsymbol{\phi}_t\}_{t \geq 0}$ generated by SPPM satisfy:*

$$\|\boldsymbol{\theta}_{t+1} - \boldsymbol{\theta}^*\|^2 + \|\boldsymbol{\phi}_{t+1} - \boldsymbol{\phi}^*\|^2 \leq \frac{\rho^2(\boldsymbol{I} - \eta\nabla_{\theta\theta}f^*)\|\boldsymbol{\theta}_t - \boldsymbol{\theta}^*\|^2 + \rho^2(\boldsymbol{I} + \eta\nabla_{\phi\phi}f^*)\|\boldsymbol{\phi}_t - \boldsymbol{\phi}^*\|^2}{1 + \eta^2\lambda_{\min}(\nabla_{\theta\phi}f^*\nabla_{\phi\theta}f^*)}$$

*where $f^* = f(\boldsymbol{\theta}^*, \boldsymbol{\phi}^*)$. Moreover, for any $\eta$ satisfying:*

$$\frac{\max(\rho^2(\boldsymbol{I} - \eta\nabla_{\theta\theta}f^*), \rho^2(\boldsymbol{I} + \eta\nabla_{\phi\phi}f^*))}{1 + \eta^2\lambda_{\min}(\nabla_{\theta\phi}f^*\nabla_{\phi\theta}f^*)} < 1, \tag{4}$$

*SPPM converges asymptotically to $(\boldsymbol{\theta}^*, \boldsymbol{\phi}^*)$.*

**Remark 5.1.** *Following the same condition as in 5.1, the iterates generated by Lv.2k GP satisfies:*

$$\|\boldsymbol{\theta}_t^{(2k)} - \boldsymbol{\theta}^*\|^2 + \|\boldsymbol{\phi}_t^{(2k)} - \boldsymbol{\phi}^*\|^2$$
$$\leq a\left(\frac{\rho^2(\boldsymbol{I} - \eta\nabla_{\theta\theta}f^*)\|\boldsymbol{\theta}_t - \boldsymbol{\theta}^*\|^2 + \rho^2(\boldsymbol{I} + \eta\nabla_{\phi\phi}f^*)\|\boldsymbol{\phi}_t - \boldsymbol{\phi}^*\|^2}{1 + \eta^2\lambda_{\min}(\nabla_{\theta\phi}f^*\nabla_{\phi\theta}f^*)}\right) + b(\|\boldsymbol{\theta}_t - \boldsymbol{\theta}^*\|^2 + \|\boldsymbol{\phi}_t - \boldsymbol{\phi}^*\|^2)$$

*where*

$$a = \frac{(1 + (\eta^2\lambda_{\max}(\nabla_{\theta\phi}f^*\nabla_{\phi\theta}f^*))^k)^2}{1 - (\eta^2\lambda_{\max}(\nabla_{\theta\phi}f^*\nabla_{\phi\theta}f^*))^k} \text{ for odd } k, \text{ or } \frac{(1 - (\eta^2\lambda_{\min}(\nabla_{\theta\phi}f^*\nabla_{\phi\theta}f^*))^k)^2}{1 - (\eta^2\lambda_{\max}(\nabla_{\theta\phi}f^*\nabla_{\phi\theta}f^*))^k} \text{ for even } k,$$
$$b = \frac{(\eta^2\lambda_{\max}(\nabla_{\theta\phi}f^*\nabla_{\phi\theta}f^*))^k(1 - (\eta^2\lambda_{\min}(\nabla_{\theta\phi}f^*\nabla_{\phi\theta}f^*))^k)}{1 - (\eta^2\lambda_{\max}(\nabla_{\theta\phi}f^*\nabla_{\phi\theta}f^*))^k}.$$

We assume that the difference between the trajectory $\{\boldsymbol{\theta}_t, \boldsymbol{\phi}_t\}_{t \geq 0}$ and the stationary point $(\boldsymbol{\theta}^*, \boldsymbol{\phi}^*)$ is not in the kernel of $\nabla_{\phi\theta} f(\boldsymbol{\theta}^*, \boldsymbol{\phi}^*)$ and $\nabla_{\theta\phi} f(\boldsymbol{\theta}^*, \boldsymbol{\phi}^*)$ for the sake of simplicity. Detailed proofs without this assumption are provided in Appendix **??**. Following the same setting as Theorem 4.1, we have $\eta < L^{-1}$, which ensures that $\eta^2\lambda_{\max}(\nabla_{\theta\phi}f^*\nabla_{\phi\theta}f^*) < 1$. Therefore, as $k \to \infty$, $a \to 1$ and $b \to 0$ in Remark 5.1, and as such Lv.$k$ GP has similar local convergence properties to SPPM. Figure 2 illustrates this property in a quadratic game. Lv.$k$ GP may behave differently than SPPM at lower values of $k$, but as $k$ increases, both max step size $\eta_{\max,k}$ and distance to equilibrium for a fixed number of iterations under $\eta_{\max,\infty}$ converges to that of SPPM. Thus, we observe that Lv.$k$ GP is empirically similar to SPPM at higher values of $k$.

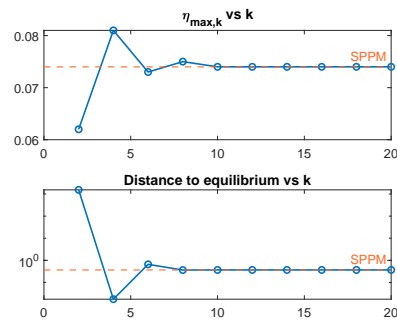

Figure 2: **Top**: max step size against $k$, **Right**: distance to equilibrium against $k$. In both figures the red dashed line represents the value for SPPM.

We study the global convergence property of SPPM by analyzing its behavior in bilinear and quadratic games. Consider the following bilinear game:

$$\min_{\boldsymbol{\theta} \in \mathbb{R}^n} \max_{\boldsymbol{\phi} \in \mathbb{R}^n} \boldsymbol{\theta}^T \boldsymbol{M} \boldsymbol{\phi} \qquad \text{(Bilinear game)}$$

where $M$ is a full rank matrix. The following theorem summarizes SPPM's convergence property:

**Theorem 5.2.** *Consider the Bilinear game and the SPPM method. Further, we define* $r_t = \|\boldsymbol{\theta}_t - \boldsymbol{\theta}^*\|^2 + \|\boldsymbol{\phi}_t - \boldsymbol{\phi}^*\|^2$. *Then, for any* $\eta > 0$, *the iterates* $\{\boldsymbol{\theta}_t, \boldsymbol{\phi}_t\}_{t \geq 0}$ *generated by SPPM satisfy*

$$r_{t+1} \leq \frac{1}{1 + \eta^2 \lambda_{\min}(\boldsymbol{M}^T \boldsymbol{M})} r_t. \tag{5}$$

It is worth noting that the convergence property of Lv.$k$ GP in bi-linear game has been studied in [2]. Furthermore, we study the convergence property of SPPM in the Quadratic game:

$$\min_{\boldsymbol{\theta} \in \mathbb{R}^n} \max_{\boldsymbol{\phi} \in \mathbb{R}^n} \boldsymbol{\theta}^T \boldsymbol{A} \boldsymbol{\theta} + \boldsymbol{\phi}^T \boldsymbol{B} \boldsymbol{\phi} + \boldsymbol{\theta}^T \boldsymbol{C} \boldsymbol{\phi} \qquad \text{(Quadratic game)}$$

where $\boldsymbol{A} \in \mathbb{R}^{n \times n}$ is symmetric and positive definite, $\boldsymbol{B} \in \mathbb{R}^{n \times n}$ is symmetric and negative definite and the interaction term $\boldsymbol{C} \in \mathbb{R}^{n \times n}$ is full rank. SPPM in quadratic games converges with the following rate:

**Theorem 5.3.** *Consider the Quadratic game and the SPPM. Then, for any* $\eta > 0$, *the iterates* $\{\boldsymbol{\theta}_t, \boldsymbol{\phi}_t\}_{t \geq 0}$ *generated by SPPM satisfy*

$$\|\boldsymbol{\theta}_{t+1} - \boldsymbol{\theta}^*\|^2 + \|\boldsymbol{\phi}_{t+1} - \boldsymbol{\phi}^*\|^2 \leq \frac{\rho^2(1 - \eta\boldsymbol{A})\|\boldsymbol{\theta}_t - \boldsymbol{\theta}^*\|^2 + \rho^2(1 + \eta\boldsymbol{B})\|\boldsymbol{\phi}_t - \boldsymbol{\phi}^*\|^2}{1 + \eta^2 \lambda_{\min}(\boldsymbol{C}^T \boldsymbol{C})} \tag{6}$$

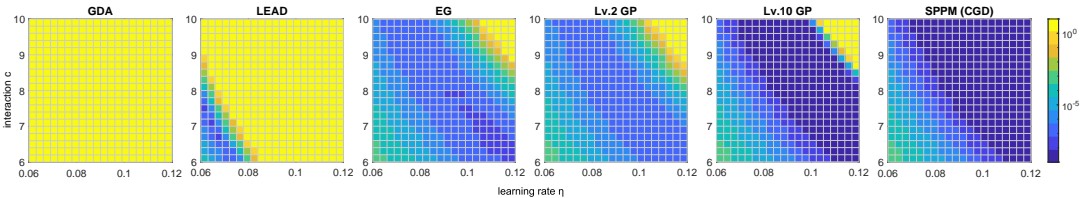

Figure 3: A grid of experiments for different algorithms with different values of interaction $c$ and learning rates $\eta$. The color in each cell indicates the distance to the equilibrium after 100 iterations. Note that the CGD update is equivalent to SPPM in quadratic games.

Theorem 5.1, 5.2 and 5.3 indicate that stronger interaction $\nabla_{\theta\phi} f(\boldsymbol{\theta}^*, \boldsymbol{\phi}^*)$ improve the convergence rate towards the stationary points. By contrast, in existing modifications of GDA, the step size is chosen in inversely proportional to the interaction term $\nabla_{\theta\phi} f(\boldsymbol{\theta}^*, \boldsymbol{\phi}^*)$ [40, 13, 34]. In Figure 3, we showcase the effect of interaction on different algorithms in the Quadratic game setup with dimension $n = 5$ and the interaction matrix is defined as $\boldsymbol{C} = c\boldsymbol{I}$. Stronger interaction corresponds to higher values of $c$. A key difference between SPPM and Lv.$k$ GP in the experiments is that, SPPM converges with any step size - and so arbitrarily fast - while it is not the case for Lv.$k$ GP. In order to approximate SPPM, one needs Lv.$k$ GP to be a contraction and so have $\eta < L^{-1}$. This result implies an additional bound on step sizes for Lv.$k$ GP. As long as the constraint on $\eta$ is satisfied, stronger interaction only improves convergence for higher order Lv.$k$ GP while all other algorithms quickly diverge as $\eta$ and $c$ increase.

## 6 Experimental Results

In this section, we discuss our implementation of Lv.$k$ GP algorithm for training GANs. Its performance is evaluated on 8-Gaussians and two representative datasets CIFAR-10 and STL-10.

### 6.1 Level $k$ Adam

We propose to combine Lv.$k$ GP with the Adam optimizer [29]. Preliminary experiments find that Lv.$k$ Adam to converge much faster than Lv.$k$ GP, see A.8 for experiment results. A detailed pseudo-code for Level $k$ Adam (Lv.$k$ Adam) on GAN training with loss functions $\mathcal{L}_G$ and $\mathcal{L}_D$ is given in Algorithm 1. For the Adam optimizer, there are several possible choices on how to update the moments. This choice can lead to different algorithms in practice. Unlike [19] where the moments are updated on the fly, in Algorithm 1, we keep the moments fixed in the reasoning steps and update it together with model parameters. In Table 2, our experiment result suggests that the proposed Lv.$k$ Adam algorithm converges asymptotically as the number of reasoning steps $k$ increases.

**Algorithm 1:** Level $k$ Adam: proposed Adam with recursive reasoning steps

---

**Input:** Stopping time $T$, reasoning steps $k$, learning rate $\eta_{\theta}, \eta_{\phi}$, decay rates for momentum
estimates $\beta_1, \beta_2$, initial weight $(\theta_0, \phi_0)$, $P_x$ and $P_z$ real and noise-data distributions,
losses $\mathcal{L}_G(\theta, \phi, x, z)$ and $\mathcal{L}_D(\theta, \phi, x, z)$, $\epsilon = 1e-8$.

**Parameters:** Initial parameters: $\theta_0, \phi_0$

Initialize first moments: $m_{\theta,0} \leftarrow 0, m_{\phi,0} \leftarrow 0$

Initialize second moments: $v_{\theta,0} \leftarrow 0, v_{\phi,0} \leftarrow 0$

**for** $t=0,\dots,T\text{-}1$ **do**

    **Sample** new mini-batch: $x, z \sim P_x, P_z$,

    $\theta_t^{(0)} \leftarrow \theta_t, \phi_t^{(0)} \leftarrow \phi_t$,

    **for** $n=1,\dots,k$ **do**

        Compute stochastic gradient:

        $g_{\theta,t}^{(n)} = \nabla_\theta \mathcal{L}_G(\theta_t, \phi_t^{(n-1)}, x, z); g_{\phi,t}^{(n)} = \nabla_\phi \mathcal{L}_D(\theta_t^{(n-1)}, \phi_t, x, z)$

        Update estimate of first moment:

        $m_{\theta,t}^{(n)} = \beta_1 m_{\theta,t-1} + (1-\beta_1)g_{\theta,t}^{(n)}; m_{\phi,t}^{(n)} = \beta_1 m_{\phi,t-1} + (1-\beta_1)g_{\phi,t}^{(n)}$

        Update estimate of second moment:

        $v_{\theta,t}^{(n)} = \beta_2 v_{\theta,t-1} + (1-\beta_2)(g_{\theta,t}^{(n)})^2; v_{\phi,t}^{(n)} = \beta_2 v_{\phi,t-1} + (1-\beta_2)(g_{\phi,t}^{(n)})^2$

        Correct the bias for the moments:

        $\hat{m}_{\theta,t}^{(n)} = \frac{m_{\theta,t}^{(n)}}{(1-\beta_1^t)}, \hat{m}_{\phi,t}^{(n)} = \frac{m_{\phi,t}^{(n)}}{(1-\beta_1^t)}; \hat{v}_{\theta,t}^{(n)} = \frac{v_{\theta,t}^{(n)}}{(1-\beta_2^t)}, \hat{v}_{\phi,t}^{(n)} = \frac{v_{\phi,t}^{(n)}}{(1-\beta_2^t)}$

        Perform Adam update: $\theta_t^{(n)} = \theta_t - \eta_\theta \frac{\hat{m}_{\theta,t}^{(n)}}{\sqrt{\hat{v}_{\theta,t}^{(n)}}+\epsilon}; \phi_t^{(n)} = \phi_t - \eta_\phi \frac{\hat{m}_{\phi,t}^{(n)}}{\sqrt{\hat{v}_{\phi,t}^{(n)}}+\epsilon}$

    $\theta_{t+1} \leftarrow \theta_t^{(k)}, \phi_{t+1} \leftarrow \phi_t^{(k)}$;

    $m_{\theta,t} \leftarrow m_{\theta,t}^{(k)}, m_{\phi,t} \leftarrow m_{\phi,t}^{(k)}$;

    $v_{\theta,t} \leftarrow v_{\theta,t}^{(k)}, v_{\phi,t} \leftarrow v_{\phi,t}^{(k)}$

---

## 6.2  8-Gaussians

In our first experiment, we evaluate Lv.$k$ Adam on generating a mixture of 8-Gaussians
with standard deviations equal to 0.05 and modes uniformly distributed around the unit cir-
cle. We use a two layer multi-layer perceptron with ReLU activations, latent dimension
of 64 and batch size of 128. The generated distribution is presented in Figure 4.

Table 2: The difference between two states generated by Lv.$k$ Adam and Lv.$k$ GP averaged over 100
steps. [†]The difference is smaller than machine precision.

| $\frac{1}{100}\sum_{t=1}^{100} r_t^{(k)}$ | $k=2$ | $k=4$ | $k=6$ | $k=8$ | $k=10$ |
|---|---|---|---|---|---|
| Lv.$k$ Adam | $1.04 \times 10^{-1}$ | $1.60 \times 10^{-2}$ | $2.91 \times 10^{-3}$ | $6.08 \times 10^{-4}$ | $1.68 \times 10^{-4}$ |
| Lv.$k$ GP | $9.79 \times 10^{-9}$ | $3.84 \times 10^{-15}$ | $1.31 \times 10^{-17}$ | $1.68 \times 10^{-19}$ | $\approx 0^{\dagger}$ |

In addition to presenting the mode coverage of the gener-
ated distribution after training, we also study the conver-
gence of the reasoning steps of Lv.$k$ Adam and Lv.$k$ GP.
Let us define the difference between the states of Lv.$k$ and
Lv.$k-1$ agents at time $t$ as:

$$r_t^{(k)} = \|\theta_t^{(k)} - \theta_t^{(k-1)}\|^2 + \|\phi_t^{(k)} - \phi_t^{(k-1)}\|^2. \quad (7)$$

We measure the difference averaged over the first 100 it-
erations, $\frac{1}{100}\sum_{t=1}^{100} r_t^k$, for Lv.10 Adam ($\eta = 10^{-4}$) and
Lv.10 GP ($\eta = 10^{-2}$). The result presented in Table2
demonstrates that both Lv.$k$ GP and Lv.$k$ Adam are con-

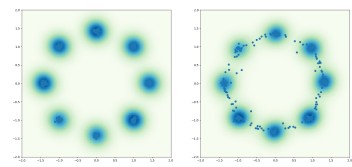

Figure 4: **Left**: real distribution, **Right**:
generated distribution

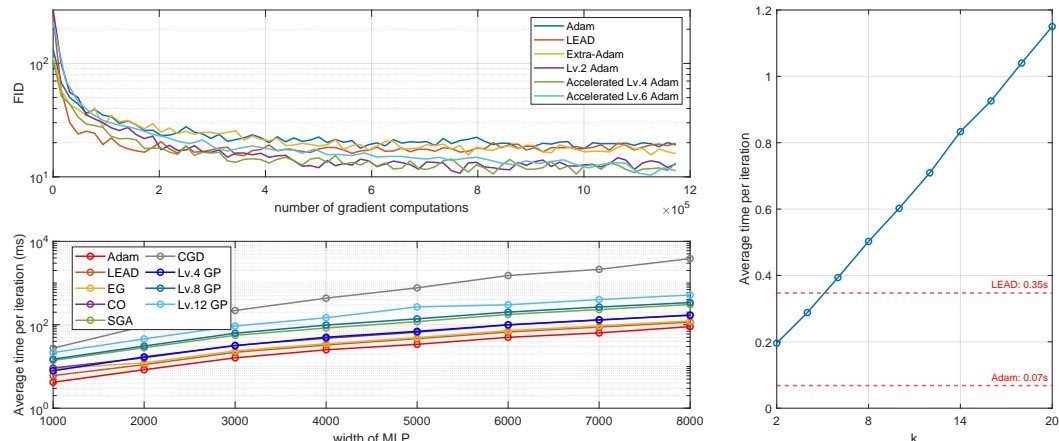

Figure 5: **Top Left**: Change of FID scores over 1.2 million gradient computations for Adam, LEAD, Extra-Adam, Lv.2, Lv.4 and Lv.6 Adam on CIFAR-10 with SNGAN. Note for Lv.4 and Lv.6 Adam, we use the accelerated implementation introduced in Appendix **??**. **Bottom Left**: Average computational cost per iteration on 8-Gaussians experiment for MLPs with different widths. **Right**: Average computational cost per iteration on CIFAR-10 for Lv.$k$-Adam with different $k$. The values for LEAD (2.88) and Adam (14.65) are highlighted by dashed line.

verging as $k$ increases. Moreover, the estimation precision of Lv.$k$ GP improves rapidly and converges to 0 within finite steps, making it an accurate estimation for SPPM.

Table 3: FID and Inception scores of different algorithms and architectures on CIFAR-10. Results are averaged over 3 runs. [†]We re-evaluate its performance on the official implementation of FID.

| Algorithm | Architecture | FID ↓ | IS ↑ |
|---|---|---|---|
| Adam [29] | BigGAN [6] | 14.73 | **9.22** |
| Adam [29] | StyleGAN2 [58] | 11.07 | 9.18 |
| Adam [29] | SN-GAN [44] | $21.70 \pm 0.21$ | $7.60 \pm 0.06$ |
| Unrolled GAN [42, 9] | SN-GAN [44] | $17.51 \pm 1.08$ | —— |
| Extra-Adam [19, 38, 9] | SN-GAN [44] | $15.47 \pm 1.82$ | —— |
| LEAD[†] [23] | SN-GAN [44] | $14.45 \pm 0.45$ | —— |
| LA-AltGAN [9] | SN-GAN [44] | $12.67 \pm 0.57$ | $8.55 \pm 0.04$ |
| ODE-GAN(RK4) [48] | SN-GAN [44] | $11.85 \pm 0.21$ | $8.61 \pm 0.06$ |
| Lv.6 Adam | SN-GAN [44] | $\mathbf{10.17 \pm 0.16}$ | $\mathbf{8.78 \pm 0.06}$ |

## 6.3  Image Generation Experiments

We evaluate the effectiveness of our Lv.$k$ Adam algorithm on unconditional generation of CIFAR-10 [31]. We use the Inception score (the higher the better) [49] and the Fréchet Inception distance (the lower the better) [24] as performance metrics for image synthesis. For architecture, we use the SN-GAN architecture based on [44]. For baselines, we compare the performance of Lv.6 Adam to that of other first-order and second-order optimization algorithms with the same SN-GAN architecture, and to state-of-the-art models trained with Adam. For Lv.$k$ Adam, we use $\beta_1 = 0$ and $\beta_2 = 0.9$ for all experiments. We use different learning rates for the generator ($\eta_\theta = 4e-5$) and the discriminator ($\eta_\phi = 2e-4$). We train Lv.$k$ Adam with batch size 128 for 600 epochs. For testing, we use an exponential moving average of the generator's parameters with averaging factor $\beta = 0.999$.

Table 4: FID scores for the different loss functions with $k = \{2, 4, 6\}$.

|                    | Lv.2 Adam        | Lv.4 Adam        | Lv.6 Adam        |
|--------------------|------------------|------------------|------------------|
| Non-saturated loss | $11.33 \pm 0.18$ | $11.62 \pm 0.25$ | $10.93 \pm 0.24$ |
| Hinge loss         | $10.68 \pm 0.20$ | $10.33 \pm 0.22$ | $10.17 \pm 0.16$ |

In table 3, we present the performance of our method and baselines. The best FID score and Inception score, $10.17 \pm 0.16$ and $8.78 \pm 0.06$, on SN-GAN are obtained with our Lv.6 Adam. We also outperform BigGAN and StyleGAN2 in terms of FID score. Notably, our model has 5.1M parameters in total, and is trained with a small batch size of 128, whereas BigGAN uses 158.3M parameters and a batch size of 2048.

**The effect of $k$ and losses:** We evaluate values of $k = \{2, 4, 6\}$ on non-saturated loss [21] (non-zero-sum formulation) and hinge loss [35] (zero-sum formulation). The result is presented in Table 4. Remarkably, our experiments demonstrate that few steps of recursive reasoning can result in significant performance gain comparing to existing GAN optimizers. The gradual improvements in the FID scores justify the idea that better estimation of opponents' next move improves performance. Moreover, we observe performance gains in both zero-sum and non-zero-sum formulations which supplement our theoretical convergence guarantees in zero-sum games.

**Experiment results on STL-10:** To test whether the proposed Lv.$k$ Adam optimizer works on higher resolution images, we evaluate its performance on the STL-10 dataset [11] with $3 \times 48 \times 48$ resolutions. In our experiments, Lv.6 Adam obtained an averaged FID of $25.43 \pm 0.18$ which outperforms that of the Adam optimizer, $30.25 \pm 0.26$, using the same SN-GAN architecture.

**Memory and computation cost:** Compared to SGD, Lv.$k$ Adam requires the same extra memory as the EG method (one additional set of parameters per player). The relative cost of one iteration versus SGD is a factor of $k$ and the computational cost increases linearly as $k$ increases, we illustrate this relationship in Figure 5 (right). We provide an accelerated version of Lv.$k$ Adam in A.8 which reduces the computation cost by half for $k > 2$. In Figure 5 (top-left), we compare the FID scores obtained by Lv.$k$ Adam, Adam, and LEAD on CIFAR-10 over the same number of gradient computations. LEAD, Lv4 Adam, and Lv6 Adam all outperform Adam in this experiment. Lv4 Adam outperforms LEAD after $2 \times 10^5$ gradient computations. Our method is also compared with different algorithms on the 8-Gaussian problem in terms of its computational cost. On the same architecture with different widths, Figure 5 (bottom-left) illustrates the wall-clock time per computation for different algorithms. We observe that the computational cost of Lv.$k$ Adam while being much lower than CGD, is similar to LEAD, SGA and CO which involve JVP operations. Each run on CIFAR-10 dataset takes $30 \sim 33$ hours on a Nvidia RTX3090 GPU. Each experiment on STL-10 takes $48 \sim 60$ hours on a Nvidia RTX3090 GPU.

## 7 Conclusion and Future Work

This paper proposes a novel algorithm: Level $k$ gradient play, capable of reasoning about players' future strategies. We achieve an average FID score of 10.17 for unconditional image generation on CIFAR-10 dataset, allowing GAN training on common computing resources to reach state-of-the-art performance. Our results suggest that Lv.$k$ GP is a flexible add-on that can be easily attached to existing GAN optimizers (e.g., Adam) and provides noticeable gains in performance and stability. In future work, we will examine the effectiveness of our approach on more complicated GAN designs, such as Progressive GANs [26] and StyleGANs [27], where optimization plays a more significant role. Additionally, we intend to examine the convergence property of Lv.$k$ GP in games with more than two players in the future.

**Broader Impact**   Our work introduces a novel optimizer that improves the performance of GANs and may reduce the amount of hyperparameter tuning required by practitioners of generative modeling. Generative models have been used to create illegal content(a.k.a. deepfakes [5]). There is risk of negative social impact resulting from malicious use of the proposed methods.

**Acknowledgement**   This work was supported by a funding from a NSERC Alliance grant and Huawei Technologies Canada. ZL would like to thank Tianshi Cao for insightful discussions on algorithm and experiment design. We are grateful to Bolin Gao and Dian Gadjov for their support.

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
