# OpenReview forum: "Recursive Reasoning in Minimax Games: A Level $k$ Gradient Play Method"
_NeurIPS.cc/2022/Conference — NeurIPS 2022 Accept_

### Official Review · Reviewer_kcBL · 2022-06-21

**Rating:** 8
**Confidence:** 3
**Soundness:** 3 good
**Presentation:** 3 good
**Contribution:** 3 good

**Summary:**

This work propose Level k Gradient Play, a new dynamical system for non-convex non-concave min-max optimization. The key feature of the dynamics is that each player tries to anticipate what the opponent will do in the following round and adapt to it instead of the opponent's current iterate. Under mild assumptions, the Level $\infty$ dynamics are well defined and enjoy local convergence for quadratic games and global convergence for bi-linear ones. In terms of practical algorithms, Level k algorithms are shown to converge to a Level $\infty$ solution as $k \to \infty$ for sufficiently small learning rates based on a contraction property. This means that they can heuristically be used as replacements for Level $\infty$. Level k Adam variants are shown to have good empirical performance.

**Questions:**

If the authors comment on the differences/similarities between this work and [1] I would increase my score. More specifically, can Level k Gradient Play or a variant thereof be considered as a special case of Clairvoyant FTRL?

**Strengths And Weaknesses:**

Regarding strengths, the presentation of the algorithm intuition and the key technical results is very clear. The proposed algorithm and analysis are to the best of my knowledge both novel when compared to other approaches in the non-convex non-concave optimization literature. While the theoretical guarantees are not particularly strong (many approaches can solve bilinear problems or have local convergence guarantees), the empirical results are promising.

The only weakness I detect is that this work is similar to [1], which is not referenced. Just like in this work, the agents try to predict the strategies of the opponents in the next turn and adapt to them. This is once again computationally feasible for small learning rates via a contraction argument. Once again the dynamics globally converge for bi-linear games and higher learning rates lead to faster convergence but may be computationally intractable just like this work.

While the overall technique may be similar to [1], the individual arguments are sufficiently different and the analysis of Section 4.1, Theorem 5.1 and 5.3 and the experimental analysis are unique to this work. Overall I propose to accept this work (Accept, 7).

I have read the response of the authors which addressed my concern. I have thus increased my score to (Strong Accept, 8).

[1] Optimal No-Regret Learning in General Games: Bounded Regret with Unbounded Step-Sizes via Clairvoyant MWU, Georgios Piliouras, Ryann Sim, Stratis Skoulakis, arXiv:2111.14737, 2021

---

> ### Author Response · Authors · 2022-08-02
> **Response to Reviewer kcBL**
>
> Dear Reviewer kcBL
>
> Thank you for your valuable feedback and suggestion, we have updated section 2 in our paper to include a brief discussion about the similarities and differences between our work and [1]. In the following section, we provide a more detailed discussion of this topic.
>
> Indeed, both Lv.k GP and clairvoyant multiplicative weights update (CMWU) incorporate the future strategy of other players in forming the agent’s current strategy, which makes both methods implicit in nature. Moreover, as noted in [1],  both methods encode in their definition a fixed point iteration, while the contraction property for a “small learning rate” is discussed in both works, [1] (v2) provides a valuable discussion about the existence of a solution for unbounded step sizes.
>
> One key difference between Lv.k GP and CMWU is their designed domain: while CMWU is designed to solve finite normal form games, Lv.k GP is designed to solve unconstrained continuous kernel games, specifically the training of GANs. This point of contrast underlies differences in settings adopted in each work. Specifically, CMWU describes the strategy profile of its agents as a simplex, whereas Lv.k GP describes the strategy profile of its agents with $R^n$. Furthermore, CMWU considers n-player general sum games, while Lv.k GP considers two-player zero-sum games. In this aspect, Lv.k GP can be viewed as a specialized variant of CMWU that is specific to the problem of two-player zero-sum games but adapted for unconstrained continuous kernel games.

---

> ### Author Response · Authors · 2022-08-07
> **Response 2 to Reviewer kcBL**
>
> Dear Reviewer kcBL,
>
> We hope we addressed your concerns with our responses and the discussion about Lv.$k$ GP and CMWU in the revised version of our paper. It would be greatly appreciated if you could reply to our responses so we still have time for potential follow-up questions. The discussion period ends on the 9th of August.
>
> Thank you so much for your time!

---

> > ### Comment · Reviewer_kcBL · 2022-08-07
> > **No additional questions**
> >
> > I acknowledge the response of the authors and I have thus increased my score to (Strong Accept, 8).

---

> > > ### Author Response · Authors · 2022-08-07
> > > **Response to Reviewer kcBL**
> > >
> > > Dear Reviewer kcBL
> > >
> > > Thank you very much. We greatly appreciate your valuable and positive feedback!

---

### Official Review · Reviewer_zfPL · 2022-07-04

**Rating:** 7
**Confidence:** 3
**Soundness:** 3 good
**Presentation:** 3 good
**Contribution:** 2 fair

**Summary:**

This paper proposes a Level K Gradient Play algorithm to stabilize the learning dynamics in minimax games (GANs).  By combining the proposed Lv. K algorithm with Adam optimizer, this  paper could achieve similar results with SOTA GAN model with 30 times fewer parameters.

**Questions:**

As stated in the **Weaknesses** section.

**Limitations:**

As stated in the **Weaknesses** section.

**Strengths And Weaknesses:**

**Strengths**:

This paper proposes the Level K Gradient Play Algorithm for stabilizing the training of GANs. The proposed method has a theoretical guarantee, with the assumption that the gradient of the loss function is Lipschitz continuous. Moreover, the paper proves and analyses the convergence properties of the Lv. K algorithm.

**Weaknesses**:
1. **On the definition of SPPM** (line 157). The author claims that "SPPM players arrive at a consensus by knowing precisely what their opponents’ future strategies will be" in line 159. However,  the stationary point $\omega^*_{t} = [\theta_t^*, \phi_t^*]$ obtained with the reasoning step in Line 139 should not equal to the future strategies $\omega_{t+1} = [\theta_{t+1}, \phi_{t+1}]$. In another word, the term $\phi_{t+1}$ used to updating $\theta_{t+1}$ does not equal to the term $\phi_{t+1}$ updated by $\theta_{t+1}$. The author should consider changing the notation here (Line 156). Or it may result in a misunderstanding that the Level.$\infty$ GP algorithm could use the opponents' future strategy to update its current gradient.

2. **Efficiency of the Level K algorithm.** As a level K algorithm would have to compute the gradient for **K** times for $\theta$ and $\phi$, this algorithm would take more time for a single step than a regular algorithm. I would recommend the author compare the method with baseline models regarding time efficiency (similar to Appendix. Figure~6 but with X axis as time).

3. **Difference with Other GAN optimizers**. The proposed methods could be seen as "given current generator, we use K step updates to find a better discriminator and use that discriminator to update the generator, and vice versa." However, some theory shows that, if we use a good (e.g., optimal) discriminator in the beginning, then we could obtain no gradient for the generator (Wasserstein-GAN). The $\omega^*$ in this paper could be seen as the optimal $\phi$ and $\theta$ with the other kept fixed. Then what is the theoretical foundation between this paper and W-GAN that makes both methods work?

4. **Limited Experiment**. This paper only conduct experiment on a small "8-Gaussians" experiment and CIFAR-10. As the author claims an improvement against BigGAN, which is good at high-resolution images, would the algorithm in this paper also be applicable to BigGAN or other big models?

5. The proposed method is to stabilize the training of GAN. However, the author also claims that this algorithm uses 30 times fewer parameters. What's the correlation between stabilizing gradients and small models? Are there any theoretical results on this issue?

---

> ### Author Response · Authors · 2022-08-02
> **Response to Reviewer zfPL**
>
> Dear Reviewer zfPL,
>
> Thank you for your valuable feedback!
>
> Before we answer your questions, we provide a detailed explanation of the reasoning mechanism in lv.k GP to serve as grounds for further discussions:
> Starting from any given state $\theta_{t},\phi_{t}$, in the first step (n=1), each player reads their opponent’s starting state, and proposes an update to their starting strategy accordingly. The updated strategy $\theta_{t}^1,\phi_{t}^1$ is shared among all players.
> $$\theta_{t}^{(1)}= \theta_{t} -\eta \nabla_{\theta}f(\theta_{t},\phi_{t}), \phi_{t}^{(1)}= \phi_{t} +\eta \nabla_{\phi}f(\theta_{t},\phi_{t})$$
> In the second step (n=2), each player reads the updated strategy of their opponent, and proposes a new update to their starting strategy according to the updated strategy of their opponent. They again share their proposed updated strategy among all players.
> $$\theta_{t}^{(2)}= \theta_{t} -\eta \nabla_{\theta}f(\theta_{t},\phi_{t}^{(1)}), \phi_{t}^{(2)}= \phi_{t} +\eta \nabla_{\phi}f(\theta_{t}^{(1)},\phi_{t})$$
> We iterate this reasoning step k times, where by step k (n=k), each player knows the updated strategy proposed by their opponent at step k-1, and proposes a final update to its starting strategy  $\theta_{t},\phi_{t}$.
> $$\theta_{t}^{(k)}= \theta_{t} -\eta \nabla_{\theta}f(\theta_{t},\phi_{t}^{(k-1)}), \phi_{t}^{(k)}= \phi_{t} +\eta \nabla_{\phi}f(\theta_{t}^{(k-1)},\phi_{t})$$
> Critically, the starting strategy of each player has not been modified, and the updates proposed in each reasoning step is not cumulative, but rather intended to be applied exclusively to the starting strategy. This distinguishes the reasoning steps from GDA, where the strategy is modified every iteration.
> To update the starting strategy after k reasoning steps, we use this last proposed strategy to modify their starting strategy, thereby obtaining $\theta_{t+1},\phi_{t+1}$.
>
>
> Regarding questions 1 and 3:
>
> We proved in Theorem 4.1 that the reasoning steps can reach a fixed point for small enough learning rate, which implies consensus between the two players. Hence, in SPPM (Lv.$\infty$ GP), the “theta updated by phi” is exactly the same as the “theta used to update phi”. This nature makes SPPM inherently implicit, similar to other implicit methods such as Proximal Point Method (PPM).
>
> We emphasize that Lv.k GP cannot be viewed or distilled to alternating optimization between two players while keeping the other player fixed. Instead, it is vital that the reasoning steps of both players occur simultaneously. In all updates, the starting point is anchored, thus they have limited search space. The \omega^* should be considered as a consensus that each player commits to play rather than an optimal generator or discriminator.
> The vanishing gradient issue of WGAN is irrelevant to our discussion, the vanishing gradient in WGAN is caused by weight clipping, which is used in the original version of WGAN to enforce Lipschitz constraints. This method has been replaced by gradient penalty and spectral normalization (the method we used in this paper) which do not suffer from this issue.
>
> Regarding question 2,
>
> Many thanks for your advice! We have provided a new set of experiments in Section 6 to demonstrate the efficiency of Lv.$k$ Adam algorithm. We compare the performance of Lv.$k$ Adam, LEAD, and Adam on CIFAR-10 over the same number of gradient computations. (each experiment takes about 30~33 hours). The experiment result indicates that Lv.$4$ Adam consistently outperforms Adam, the baseline model, in terms of FID score and it starts to outperform LEAD after $2\times10^5$ gradient computations. We also provide a figure showing how the computational time increases as k increases. In practice, Lv.$k$ Adam has similar cost to methods based on Jacobian Vector Product operations (CO, SGA, LEAD).
>
> Regarding question 4
>
> Our computation budget does not allow us to get such results*, but we have included a set of experiments using STL-10 dataset, which has a higher resolution (3X48X48) than CIFAR-10.
>
> *BigGAN was trained on 8xV100 and it takes 15 days to train to 150k iterations (referring to resolution of 128x128 pixels), the cost is approximately $5,000 on google cloud per experiment.
>
> Regarding question 5
>
> We are sorry if the statement in our paper is misleading, we mention that the architecture (SNGAN) uses 30 times fewer parameters than BigGAN to emphasize that improvements on the optimizer is as important as advancement in model architectures. A simple model with a good optimizer can beat complex models with baseline optimizers. In our last revised version, Remark 5.1 provides further insight into the local convergence properties of Lv.k GP, which when combined with the compelling empirical results increases the likelihood that performance will follow.

---

> > ### Comment · Reviewer_zfPL · 2022-08-03
> > **Comments**
> >
> > Thanks for the classification. I make a mistake when reading the equation (Lv.k GP), as $ \theta_t^{(n)} = \theta_t - \eta \nabla_{\theta}f((\theta_t^{(n-1)}, \phi_t) $. Now I have no problem on question 1 and question 3. Thanks.

---

> > > ### Author Response · Authors · 2022-08-03
> > > **Response to Reviewer zfPL**
> > >
> > > Dear Reviewer zfPL,
> > >
> > > We appreciate your quick response. Please note that we also answered questions 2, 4, and 5 in your official review. Please let us know if you have further questions. We would appreciate it if you could give us a higher rating if our response resolved your concerns.

---

> > > ### Author Response · Authors · 2022-08-07
> > > **Response 2 to reviewer zfPL**
> > >
> > > Dear Reviewer zfPL,
> > >
> > > We hope we have addressed your concerns with our responses and revisions in our paper. We would greatly appreciate it if you could respond as soon as possible if you have any further comments. If our responses addressed your concerns, please consider giving us a higher rating. The discussion period ends on the 9th of August.
> > >
> > > Thank you so much for your time.

---

### Official Review · Reviewer_Rboc · 2022-07-11

**Rating:** 4
**Confidence:** 3
**Soundness:** 2 fair
**Presentation:** 2 fair
**Contribution:** 3 good

**Summary:**

This work introduces a novel algorithm, closely related to existing "lookahead approaches", for solving minmax games.
In "level k gradient play" each agent first arrives at a prediction of the opponent's move through k steps of recursive reasoning  *the counter move to the counter move to ... gradient descent*. They then make a step of gradient descent in the direction of the gradient computed using their own present position and their opponent's position according to the recursive reasoning. The authors show convergence results for the (theoretical) infinite recursion version with $k = \infty$. Finally, the authors show numerical experiments on GANs, demonstrating improved IS and FID scores.

**Questions:**

1. I find the way in which the algorithms are studied through the lens of SPPM to be confusing. First (up to including section 4), different algorithms are compared regarding how good they are at approximating SPPM. To this point, only vague, anthropomorphic language is provided to justify why approximating SPPM should be desirable. For instance in line 159: "SPPM players arrive at a consensus by knowing precisely what their opponents’ future strategies will be. Existing algorithms are not able to offer this kind of agreement."
In Section 5, justification is given why SPPMs could be interesting by providing convergence results of SPPM. It is also claimed (line 189), that this convergence result is going to be used to study the convergence properties of Lv.k GP. However, this is then declared "beyond the scope of this work" and it is claimed that the convergence property can be "inferred ... via SPPM", without providing rigorous definitions.
In my opinion, this work could be strengthened considerably if instead of the SPPM characterization a simple local convergence result of Lv.k GP would be given. At first glance, it seems that it should be possible (strictly convex problem, small enough step size), that local convergence can be essentially reduced to the quadratic problem, where Lv.k GP should not be so hard to characterize? I would be curious to hear the opinion of the authors as to what is the roadblock here. Even if an analysis of Lv.k GP is not possible, it seems that by combining the theorems already in the paper it should be possible to provide an analysis for Lv.inf GP.

2. The authors advertise their methods as not requiring second-order gradient information, compared to other methods that ...
"demand second-order gradient information (e.g., Jacobian-vector product and Hessian inverse), making well-performing models unavailable for standard computational budgets [50, 6]"
This claim is false or at least misleading, as the proposed methods compute gradients at multiple points at each iteration and thus also use second-order information. Furthermore, using performant mixed-mode automatic differentiation frameworks such as JAX, computing Jacobian-vector products need not be significantly more expensive than gradient computation (for Hessian-inverses the story is a bit more complicated). Thus, the above strikes me as mischaracterizing competing methods.

3. Related to the above, none of the experimental data shown give an idea of the cost-vs-quality trade-off of the proposed method. Can you run, say, Adam and Lv.6 Adam side by side and plot the IS and FID as a function of the number of gradient computations?

4. Just out of curiosity: Did you try what happens if you run the experiment shown in Table 2 on the larger, image datasets?

5. You write in line 164 that "In games with at least thrice differentiable objective function". Since the bilinear objective is infinitely differentiable you probably meant something else. Similar in 176.

6. Is there a reason why LOLA and CGD were not included in Figure 2?

**Limitations:**

The authors discuss the additional computational cost of Lv.k in the very end of section 6. However, I was surprised that they did not provide further investigation of this concern by comparing, for instance, and adam model with k times as many epochs to an Lv.k Adam.

**Strengths And Weaknesses:**

### Strengths:

- The algorithm seems natural and well-motivated to me.
- The algorithm appears to provide substantial improvements in GAN training

### Weaknesses:

- The emphasis on SPPM is confusing and makes it hard to understand how the proposed algorithm relates to other methods in the literature.
- Overall, there is a lot of hand-wavey language in the paper that hints at the advantages of the proposed method or results of the paper, that are never made concrete.

Overall, my present view of the paper is that the results are suitable to be published in NeurIPS. However, the paper needs to undergo a major reorganization/rewrite to clearly convey its key points. I am not sure such a revision is within the scope of a single conference review cycle which is why I tend towards rejection and encouraging the authors to resubmit to a future cycle.

---

> ### Author Response · Authors · 2022-08-02
> **Response to Reviewer Rboc**
>
> Dear Reviewer Rboc,
>
> Many thanks for your extensive review and detailed comments. Below we answer separately to your questions.
>
> Regarding question 1,
>
> Thank you for your suggestion on how to strengthen our paper. In our revised paper, the local convergence property of Lv.k GP is no longer “beyond the scope of this work” as we characterize the local convergence property of Lv.k GP in Remark 5.1 in terms of squared Euclidean norm in the parameter space. The local convergence bound presented in Remark 5.1 contains problem-dependent terms for specific $k$, but we prove that its behavior is strongly correlated to that of SPPM - specifically, the local convergence bound of lv.k GP can be formulated as a scaled and shifted version of the same bound for SPPM. As expected, the difference between Lv.k GP and SPPM in terms of local convergence property will converge to 0 as $k$ increases for a small enough learning rate. In addition, we provide a set of experiment results on the quadratic game to demonstrate that Lv.k GP is empirically similar to SPPM at higher values of k.
>
> Regarding questions 2 and 3,
>
> We are sorry for the confusion regarding computational efficiency.  Our model relies only on gradient information, which has a similar cost to JVP operations. For a fair comparison with our methods, we account for each JVP operation as one gradient computation. (JVP method has a similar cost to our method, our method is faster than CGD which involves Hessian inverse and conjugate gradient in practice.)
>
> To study the performance gain of Lv.k Adam, we compare the FID score obtained by Lv.k Adam, LEAD, and Adam over 1.2 million gradient computations, the result is presented in Figure 5 (top left). Note we also tried SGA but it diverged at an early stage thus it is not included in Figure 5. As indicated by the experiment result, Lv 4 Adam consistently outperforms LEAD after $2\times 10^5$ gradient computations.
>
> To compare the computational cost of our method with algorithms that involve Hessian inverse and JVP operations, we measure the average time per iteration on 8-Gaussians tasks for MLPs with different widths. The result is presented in Figure 5 (bottome left). The result indicates that while Lv.k GP has similar computational cost to algorithms that involve JVP operations, it is much faster than algorithms that require Hessian inverse operations (which is approximated by conjugate gradient in our implementation) in its update.
>
> In Figure 5 (right), we also illustrate how does the computational time increase as k increases. It is worth noting that, both Lv.4 Adam and LEAD involves 4 gradient compurations but Lv.4 Adam is slightly faster than LEAD which may account for the difference between gradient computation and JVP operation.
>
> Regarding the remaining questions,
>
> Thank you for pointing out our misusage of terminologies, we have rewritten the corresponding sentences in Section 4 as
> “in games with cost functions that have non-negative higher order derivatives $n \geq 3$”. The reason for this clarification is that CGD is equivalent to SPPM in bilinear and quadratic games. We have updated Figure 3 to notice that the plot for SPPM also accounts for the performance of CGD. Moreover, we also include a more detailed plot in Appendix A.7 which includes the experiment result for LOLA (SGA) and more Lv.k GPs.

---

> > ### Comment · Reviewer_Rboc · 2022-08-07
> > **Thanks for addressing my comments**
> >
> > Thank you for the detailed response to my concerns and for improving the paper. In light of the new results (both theoretically and empirically), could you comment on the benefits of LV.k compared to a simple extragradient approach? At first glance, the errors in convergence rates in remark 5.1 seem to behave like $\approx c^(tk)$. Since the cost per timestep is $\approx k$, it is not clear to me what the benefit of lv.k is?
> > The same pattern is visible in the new experimental results, where Lv.6 performs worse than Lv.4, suggesting that extragradient might perform better than both of those methods. I thank the authors for reporting the additional results that have the potential of filling a gap in the literature but I am presently a bit at a loss about the motivation for using Lv.k GP.

---

> > > ### Author Response · Authors · 2022-08-08
> > > **Thank you for your comments!**
> > >
> > > Dear Reviewer Rboc,
> > >
> > > Thank you for your valuable feedback.
> > >
> > > We should note that Figure 5 (top left) aims to compare the efficiency of Lv.$4$ Adam and Lv.$6$ Adam, considering the difference between the number of epochs (500 vs 750), it does not indicate that Lv.$4$ Adam outperforms Lv.$6$ Adam in terms of best FID scores.
> > >
> > > In fact, in Table 3, we compared the best-reported FID scores obtained with Extra-Adam and Lv.6 Adam, in Table 4 we also provide the empirical results for Lv.$2$ Adam and Lv.$4$ Adam on the same task. The experiment result indicates that Lv.$k$ Adam consistently outperforms Extra-Adam in terms of best FID scores, moreover, Lv.$6$ Adam performs better than Lv.$2$ and Lv.$4$ Adam.
> > >
> > > Considering the efficiency of different algorithms, if we use the accelerated Lv.$k$ Adam given in Appendix A.10, Lv.$4$ Adam requires the same number of gradient computations as extragradient method:
> > >
> > > Accelerated Lv$4$ GP:
> > > $$x_{t}^{(1)} = x_{t} - \nabla_{x}f(x_{t},y_{t}^{(0)}); y_{t}^{(2)} = y_{t} + \nabla_{y}f(x_{t}^{(1)},y_{t}).$$
> > > $$x_{t}^{(3)} = x_{t} - \nabla_{x}f(x_{t},y_{t}^{(2)}); y_{t}^{(4)} = y_{t} + \nabla_{y}f(x_{t}^{(3)},y_{t}).$$
> > > whereas
> > > extragradient:
> > > $$x_{t}^{(1)} = x_{t} - \nabla_{x}f(x_{t},y_{t}); y_{t}^{(1)} = y_{t} + \nabla_{y}f(x_{t},y_{t}).$$
> > > $$x_{t}^{(2)} = x_{t} - \nabla_{x}f(x_{t}^{(1)},y_{t}^{(1)}); y_{t}^{(2)} = y_{t} + \nabla_{y}f(x_{t}^{(1)},y_{t}^{(1)}).$$
> > >
> > > (Accelerated Lv.$2$ GP is the same as alternating gradient descent ascent.)
> > >
> > > Regarding the performance gain and computational cost, we believe Lv.$k$ Adam is a better choice than extragradient method for GANs.

---

> > > > ### Comment · Reviewer_Rboc · 2022-08-08
> > > > **Thanks for comments**
> > > >
> > > > For the theory part of my question: Does the theory cover accelerated lv.$k$ adam / GP or the "exact" one that takes $k$ gradient computations?
> > > > Regarding the experiments: What reference is the extra-Adam FID coming from? I checked the first reference provided, and it does not seem to report any FID scores.

---

> > > > > ### Author Response · Authors · 2022-08-08
> > > > > **Thank you for your comment.**
> > > > >
> > > > > Dear Reviewer Rboc
> > > > >
> > > > > Thank you very much for your feedback!
> > > > >
> > > > > Regarding question 1,
> > > > >
> > > > > The short answer is no. The theory is derived for Algorithm 1, non-accelerated Lv.$k$ GP which takes $2k$ gradient computations. The experiment results in Figure 5 we reported are using accelerated Lv.$k$ Adam. However, the updates in accelerated Lv.$k$ GP are exactly the same as Lv.$k$ GP except for the alternating order.
> > > > >
> > > > > Let us again consider accelerated Lv.$4$ GP as an example
> > > > >
> > > > > Accelerated Lv$4$ GP:
> > > > > $$x^{(1)} = x_{t} - \nabla_{x}f(x_{t},y_{t}^{(0)}); y^{(2)} = y_{t} + \nabla_{y}f(x_{t}^{(1)},y_{t}).$$
> > > > > $$x^{(3)} = x_{t} - \nabla_{x}f(x_{t},y_{t}^{(2)}); y^{(4)} = y_{t} + \nabla_{y}f(x_{t}^{(3)},y_{t}).$$
> > > > >
> > > > > The state $x^{(3)}$ is exactly the same as the player $x$ in (non-accelerated) Lv.$3$ GP and $y^{(4)}$ is the same as $y$ in (non-accelerated) Lv.$4$ GP. Therefore, we would expect them to follow a similar convergence property as proved in Remark 1.
> > > > >
> > > > > Regarding question 2,
> > > > >
> > > > > The reported FID can be found in Table 1 of [9]. The reason we also cited [19] and [38] is because they first introduced extra-adam. We will make a clarification in our paper if this is misleading.
> > > > >
> > > > > [9] Tatjana Chavdarova, Matteo Pagliardini, Sebastian U Stich, François Fleuret, and Martin Jaggi.
> > > > > Taming GANs with lookahead-minmax. arXiv preprint arXiv:2006.14567, 2020.

---

> > > > > > ### Comment · Reviewer_Rboc · 2022-08-08
> > > > > > **Thanks for the response**
> > > > > >
> > > > > > Thanks for the response. Would it be possible to simply add a line showing the results on extragradient to the top left plot of Figure 5? Given how closely related this method is to the proposed methods, I find it puzzling why this comparison was not shown and instead the much less commonly used LEAD was used as a comparison.

---

> > > > > > > ### Author Response · Authors · 2022-08-08
> > > > > > > **Thank you for your comments**
> > > > > > >
> > > > > > > Dear Reviewer Rboc,
> > > > > > >
> > > > > > > Thank you very much for your suggestion. We updated the paper and include a line showing the results of Extra-Adam on CIFAR-10 in the top left plot of Figure 5.
> > > > > > >
> > > > > > > We include the results for LEAD in Figure 5 since it involves Jacobian-vector-product operation, we are trying to compare the efficiency of our method with algorithms based on JVP operations (in response to Q2 of your initial comments). We promise that we will include a set of experiment results of Extra-Adam in Figure 5 in the camera-ready version if our paper is accepted.
> > > > > > >
> > > > > > > Thank you very much for your time!

---

> > > > > > > > ### Comment · Reviewer_Rboc · 2022-08-08
> > > > > > > > **Sorry I was misunderstood**
> > > > > > > >
> > > > > > > > Sorry I did not mean adding a line representing the reported results from elsewhere, but running the extra-adam comparison. Given that Lv.4 performs better than Lv.6, it seems worthwhile to further lower the number of recursive reasoning steps and, in a sense, extra gradient corresponds to just a single additional reasoning step.

---

> > > > > > > > > ### Author Response · Authors · 2022-08-08
> > > > > > > > > **Thanks for your feedback**
> > > > > > > > >
> > > > > > > > > Thank you for your response.
> > > > > > > > >
> > > > > > > > > We fully understand your concern. But each experiment takes about 30~33 hours, regarding the time concerns, we are not able to include this experiment in this discussion phase. As we promised, we will include this experiment result in Figure 5 in the camera-ready version if our paper is accepted.
> > > > > > > > >
> > > > > > > > > We would like to argue that, compared to the experiment results in [9], we are using exactly the same model (SNGAN), the same set of hyperparameters, and both uses the adaptive version of the original algorithm, we believe that the comparison is fair and convincing.
> > > > > > > > >
> > > > > > > > > Additionally, a set of experiment results for Lv.$2$ Adam will also be included as it corresponds to the case with just a single additional reasoning step although it in fact requires the same number of gradient computations as the accelerated Lv.$4$ Adam.

---

> > > > > > > > > ### Author Response · Authors · 2022-08-08
> > > > > > > > > **Further Response**
> > > > > > > > >
> > > > > > > > > We would like to note that Figure 5 only compares the efficiency of Lv.$4$ Adam and Lv.$6$ Adam, it does not indicate that the best FID score (which is one of the most important metrics for evaluating GANs) obtained by Lv.$4$ Adam (10.68) is better than Lv.$6$ Adam (10.33). Moreover, in Figure 5, Lv.$6$ Adam is only trained for 500 epochs while Lv.$4$ Adam is trained for 750 epochs. Table 4 also demonstrates the difference in terms of the best FID score obtained by Lv.$k$ Adams with different values of $k$. Considering all aspects, we cannot simply conclude that Lv.$4$ is better than Lv.$6$.

---

> > > > > > > > > ### Author Response · Authors · 2022-08-08
> > > > > > > > > **New experiment results for Lv.$2$ Adam**
> > > > > > > > >
> > > > > > > > > Dear Reviewer Rboc,
> > > > > > > > >
> > > > > > > > > We have now included a new set of experiment results for Lv.$2$ Adam in the top left plot of Figure 5. The data points are collected from the experiment results reported in Table 4. As mentioned in the previous discussion, Lv.$2$ Adam requires the same number of gradient computations per iteration as accelerated Lv.$4$ Adam and the result indicates that accelerated Lv.$4$ Adam improves slightly faster than Lv.$2$ Adam in the early stage. The best FID score for Lv.$2$ Adam is 10.76, for accelerated Lv.$4$ Adam is 10.68, and for accelerated Lv.$6$ Adam is 10.33. We hope this response can address your concern about the missing experiment.
> > > > > > > > >
> > > > > > > > > Thank you for your time!

---

> > > > > > > > > ### Author Response · Authors · 2022-08-09
> > > > > > > > > **Final best attempt. We provide new experiment results in Figure 5.**
> > > > > > > > >
> > > > > > > > > Dear Reviewer Rboc
> > > > > > > > >
> > > > > > > > > On the one hand, we really appreciate all the detailed feedback and comments. Your consistent communication with us has been very helpful to us. Your comments definitely strengthened our paper. On the other hand, we are a bit frustrated and upset. We believe we have addressed all your concerns so far. We cannot participate further in the reviewer discussion period, which is a bit unfair to us. However, we won't give up and would like to give it our final best try. We hope to reach a consensus before the rebuttal deadline.
> > > > > > > > >
> > > > > > > > > In your last response, you raised concerns about the comparisons between the extra-Adam algorithm and the algorithm that involve fewer reasoning steps. We stress again that we have presented the best FID score obtained with Extra-Adam in Table 3. According to Appendix H of [9] we are using exactly the same model SNGAN, same batch size, and similar hyperparameters. Both methods are fine-tuned to best behavior. We believe this result is convincing and it is easy to infer that in the worst case, Lv.$4$ Adam outperforms Extra-Adam after $2\times 10^{5}$ gradient computations. (Since the best FID score for Extra-Adam is 15.47, similar to 15.39 obtained by LEAD in Figure 5).
> > > > > > > > >
> > > > > > > > > As for experiments with fewer reasoning steps, we provide a new set of experiment results for Lv.$2$ Adam in the top left plot of Figure 5. It is not better than the accelerated Lv.$4$ Adam which requires exactly the same number of gradient computations per iteration.
> > > > > > > > >
> > > > > > > > > We believe our paper deserves a higher rating given our theoretical and empirical contributions. We've resolved all your concerns so far, and we hope you can reconsider your rating.
> > > > > > > > >
> > > > > > > > > Thank you very much for your time and suggestions.

---

> > > > > > > > > > ### Comment · Reviewer_Rboc · 2022-08-09
> > > > > > > > > > **Thank you for your patience**
> > > > > > > > > >
> > > > > > > > > > I'm aware that the discussion period is running out and understand that this is frustrating from an author's perspective. Please rest assured that I am not trying to "play for time" but am doing my best to facilitate a productive discussion. However, I have other responsibilities that prevent me from responding as quickly as I would like to.
> > > > > > > > > >
> > > > > > > > > > I appreciate you adding the new results for Lv.2 Adam, despite them not supporting the case of the paper. Since, to my understanding, Lv.2 Adam amounts to applying the extra gradient procedure to alternating as opposed to simultaneous gradient descent, I am still concerned about the experimental results. From the experiments on GANs provided by the authors, there seems to be no benefit from the deeper recursive reasoning when fairly accounting for the number of gradient computations performed. A similar conclusion is implied by Figure 9 of the appendix. Since these results oppose the core tenets of the paper, I feel uneasy about accepting it at this state.
> > > > > > > > > >
> > > > > > > > > > On the other hand, I appreciate that not every paper has to "win" at benchmarks and believe that a thorough study of the effect of Lv.k type algorithms, as provided by the authors, may merit a place in the literature independent of its conclusions. To this end, I greatly appreciate the authors' efforts to make the theoretical picture more complete.
> > > > > > > > > >
> > > > > > > > > > Overall, I am undecided about what the best decision is on this paper and look forward to discussing it with the other reviewers and the AC.

---

> > > > > > > > > > > ### Author Response · Authors · 2022-08-09
> > > > > > > > > > > **Thank you for your comments and valuable suggestions.**
> > > > > > > > > > >
> > > > > > > > > > > Dear Reviewer Rboc,
> > > > > > > > > > >
> > > > > > > > > > > As participants of Neurips 2022, we sincerely thank you for your candid comments and helpful suggestions. We really appreciate your consistent communication with us. We believe that you have done a great job as a reviewer.
> > > > > > > > > > >
> > > > > > > > > > > As the author of this paper, we will try our best to address your concerns.
> > > > > > > > > > >
> > > > > > > > > > > First, we stress that Lv.$2$ GP is different from the official Extra-gradient method in the existing literature [1].
> > > > > > > > > > >
> > > > > > > > > > > Lv$2$ GP:
> > > > > > > > > > > $$x^{(1)} = x_{t} - \eta\nabla_{x}f(x_{t},y_{t}); y^{(1)} = y_{t} + \eta\nabla_{y}f(x_{t},y_{t}).$$
> > > > > > > > > > > $$x^{(2)} = x_{t} - \eta\nabla_{x}f(x_{t},y_{t}^{(1)}); y^{(2)} = y_{t} + \eta\nabla_{y}f(x_{t}^{(1)},y_{t}).$$
> > > > > > > > > > > whereas
> > > > > > > > > > > Extra-gradient:
> > > > > > > > > > > $$x^{(1)} = x_{t} - \eta\nabla_{x}f(x_{t},y_{t}); y^{(1)} = y_{t} + \eta\nabla_{y}f(x_{t},y_{t}).$$
> > > > > > > > > > > $$x^{(2)} = x_{t} - \eta\nabla_{x}f(x_{t}^{(1)},y_{t}^{(1)}); y^{(2)} = y_{t} + \eta\nabla_{y}f(x_{t}^{(1)},y_{t}^{(1)}).$$
> > > > > > > > > > >
> > > > > > > > > > > From the perspective provided by [2], the Extra-gradient method can be viewed as an approximation of the Proximal Point Method while Lv.$2$ Adam approximates the Semi-Proximal Point Method we introduced in Section 4 of our paper. Thus, Lv.$2$ Adam is not the Extra-gradient method or a variant of the Extra-gradient method, it is a novel algorithm that has not been used in any previous works on GANs.
> > > > > > > > > > >
> > > > > > > > > > > We agree that a comprehensive study on the difference between the Extra-gradient method and Lv.$2$ GP algorithm (i.e. the difference between PPM and SPPM) in non-convex games can strengthen our paper, but regarding the difficulty of analyzing optimization in non-convex games, we are not able to include a rigorous analyzation within the scope of this paper.
> > > > > > > > > > >
> > > > > > > > > > > As for the experiment result, we insist that FID vs the number of gradient computations is a metric used for evaluating the $\textbf{efficiency}$ of optimizers, but not the $\textbf{best FID}$ that can be attained by an optimizer. Although efficiency is a very important metric for the evaluation of optimizers, the number of gradient computations is in fact not a bottleneck for many practical GAN applications. Moreover, since Lv.$k$ Adam allows for different numbers of reasoning steps, it is possible to train the model at an early stage using fewer $k$ for better efficiency and then gradually increase $k$ for a higher FID score. Table 3 and Table 4 justify the fact that Lv.$k$ Adam is by far the optimizer that obtains the best FID score on CIFAR-10 using SNGAN. We believe this is strong enough to support our claim that we have introduced a novel and effective optimizer for solving minimax games.
> > > > > > > > > > >
> > > > > > > > > > > We would like to sincerely thank you again for your consistent communication and help. The valuable discussion with you helped us a lot to strengthen our paper.
> > > > > > > > > > >
> > > > > > > > > > > [1] Korpelevich, G. The extragradient method for finding saddle points and other problems. Matecon,
> > > > > > > > > > > 12:747–756, 1976.
> > > > > > > > > > >
> > > > > > > > > > > [2] Aryan Mokhtari, Asuman Ozdaglar, and Sarath Pattathil. A unified analysis of extra-gradient and optimistic gradient methods for saddle point problems: Proximal point approach. In International Conference on Artificial Intelligence and Statistics, pages 1497–1507. PMLR, 2020.

---

> > > > > ### Author Response · Authors · 2022-08-08
> > > > > **A brief summary of our responses to your concerns**
> > > > >
> > > > > Dear Reviewer Rboc,
> > > > >
> > > > > We thank you for all the detailed feedback and comments. We really appreciate your consistent communication with us. Your comments definitely helped make our paper stronger. As we believe we have addressed all your concerns, here is a brief summary in case you have any further questions.
> > > > >
> > > > > >In light of the new results (both theoretically and empirically), could you comment on the benefits of LV.k compared to a simple extragradient approach?   I thank the authors for reporting the additional results that have the potential of filling a gap in the literature but I am presently a bit at a loss about the motivation for using Lv.k GP.
> > > > >
> > > > > First, Lv.2 GP is different from extra-gradient method. Using the empirical result provided in Table 3 and Table 4, we demonstrate that Lv.2 Adam, the adaptive version of Lv.2 GP significantly outperforms Extra-Adam, the adaptive version of extra-gradient method. Moreover, we justified that accelerated Lv.4 GP requires the same number of gradient computations as extra-gradient method. Therefore, Lv.k GP outperforms extragradient method in both performance and efficiency.
> > > > >
> > > > > >For the theory part of my question: Does the theory cover accelerated lv.k adam / GP or the "exact" one that takes k gradient computations?
> > > > >
> > > > > We prove that the updates of accelerated Lv.k GP are identical to Lv.k GP except for the alternating orders. Both are approximations of SPPM, thus they should follow similar convergence properties as provided in Remark 5.1.
> > > > >
> > > > > >Regarding the experiments: What reference is the extra-Adam FID coming from? I checked the first reference provided, and it does not seem to report any FID scores.
> > > > >
> > > > > We use the result provided in [9], which by far is the best FID reported for Extra-Adam on CIFAR-10 using SNGANs. We have provided a copy of our model trained with Lv.6 Adam using the same model and a test script in the supplementary material for your reference.
> > > > >
> > > > > Please do not hesitate to discuss with us if you have any unaddressed concerns. We believe our paper deserves a higher score given our theoretical and empirical contributions. If we have resolved your concerns, we hope you can reconsider your rating.

---

> ### Author Response · Authors · 2022-08-07
> **Response 2 to Reviewer Rboc**
>
> Dear Reviewer Rboc
>
> We thank you for your extensive review and detailed comments. We hope we addressed your concerns with our responses and the new results we included in our paper. We would greatly appreciate it if you could reply to our responses as soon as possible so that we still have time to discuss follow-up questions. The discussion period ends on the 9th of August.
>
> Thank you so much for your time.

---

### Official Review · Reviewer_iDC6 · 2022-07-14

**Rating:** 5
**Confidence:** 3
**Soundness:** 3 good
**Presentation:** 4 excellent
**Contribution:** 3 good

**Summary:**

This paper studied optimization of minimax games and proposed a recursive optimization algorithm called Level k gradient play (Lv.k GP). As an update based on predictive updates, Lv.k GP does not require second-order information, which is computationally more efficient than existing algorithms.

Theorem 4.1 showed that as k increases, the level k reasoning of parameter vectors is approaching a limit point. And Lv.$\infty$ GP is equivalent to an ideal algorithm Semi-Proximal Point Method (SPPM).

Table 1 then summarized different algorithms and viewed them as approximations of SPPM, and Lv.k GP approximation accuracy improves as k increases.

Theorem 5.1 showed local convergence of SPPM toward stationary points, and Theorems 5.2 and 5.3 showed that for a specific bilinear game and quadratic game, the SPPM converges in terms of squared norm of parameter distances.

Experiments on training GANs using 8-Gaussians, CIFAR-10 and STL-10 are conducted to show that Lv.k GP and Lv.k Adam can be used to help existing GAN optimizers (e.g., Adam) provides noticeable gains in performance and stability.

**Questions:**

1. It is claimed that the Lv.k GP methods do not require second order information. Thus I suppose the implication is that Lv.k is faster than second order methods and their first order approximations (e.g., rank-1 approximation)? Is there any evidence to support this?

2. How does the computational time increase as k increases (seems this relation should be linear to me)? It would be better to have the computational time recorded.

3. Theorems 5.1-5.3 studied the convergence properties of the ideal algorithm SPPM. However, the algorithms actually used here are Lv.k GPs. A more direct study should be to get convergence properties of Lv.k GP, where as $k \to \infty$, Theorems 5.1-5.3 could be recovered.

**Limitations:**

The authors discussed the potential negative impact of using GANs to generate images, which is reasonable to me.

**Strengths And Weaknesses:**

Strengths:

1. Minimax game optimization is an important problem which has lots of applications such as GANs, while being difficult to resolve and optimize. The research is well motivated.

2. The proposed Lv.k GP algorithm seems novel to me, and the authors provided its approximation to SPPM and showed the convergence properties of SPPM.

3. Experimental results are supporting the proposed methods

Weakness:

1. The proposed algorithms are Lv.k GPs, while the convergence properties are studied for SPPM method, which creates a gap.

2. It is not clear that whether the claim of Lv.k GP is faster than (first order approximations) of second order methods is true or not.

---

> ### Author Response · Authors · 2022-08-02
> **Response to Reviewer iDC6**
>
> Dear Reviewer iDC6
>
> Thank you for your valuable feedback. We will try to address your concerns in the following paragraph.
>
> Regarding questions 1 and 2:
>
> We clarify in section 1, that our method is faster than second-order methods that involve Hessian inverse operations (CGD and Follow the ridge). Meanwhile, our method outperforms JVP-based methods in terms of FID score. In section 6, we provide a new set of experiments to support our claim, with results illustrated in Figure 5.
>
> In Figure 5 (top left), we compare the FID score obtained by Lv.k Adam, LEAD, and Adam over 1.2 million gradient computations. Note we also tried SGA but it diverges at an early stage thus it is not included in Figure 5. The experiment result indicates that after $2\times 10^5$ gradient computations, Lv.4 Adam consistently outperforms LEAD.
>
> In Figure 5 (right), we also illustrate how the computational time increases as k increases. It is worth noting that both Lv.4 Adam and LEAD involve 4 gradient computations, but Lv.4 Adam is slightly faster than LEAD, which is likely due to the difference between gradient computation and JVP operation.
>
> As for Hessian inverse based CGD (which is approximated by conjugate gradient in our implementation), we compare its average time cost per iteration for different model sizes with other algorithms and present the results in Figure 5 (bottom left). The result indicates that while Lv.k GP has a similar computational cost to algorithms that involve JVP operations, it is much faster than algorithms that require Hessian inverse operations in its update.
>
> Regarding question 3:
>
> We provide Remark 5.1 in our revised paper to study the local convergence property of Lv.k GP in non-convex non-concave games. Remark 5.1 characterizes the local convergence property of Lv.k GP in terms of the squared Euclidean norm in the parameter space. Although the local convergence property for specific $k$ is problem-dependent, its behavior shows a strong correlation to that of SPPM. As expected, the difference between Lv.k GP and SPPM in terms of local convergence property will converge to 0 as $k$ increases for a small enough learning rate. In addition, we provide a set of experimental results on the quadratic game to demonstrate that Lv.k GP is empirically similar to SPPM at higher values of k in Figure 2.

---

> > ### Author Response · Authors · 2022-08-07
> > **Response 2 to Reviewer iDC6**
> >
> > Dear Reviewer iDC6
> >
> > We hope our responses and the corresponding changes made to the paper can address your concerns about
> > 1. the theoretical connections between Lv.$k$ GP and SPPM,
> > 2. the empirical comparison of Lv.$k$ Adam with baseline models.
> >
> > We would greatly appreciate it if you could reply to our responses as soon as possible so we can continue to discuss follow-up questions. The discussion period ends on the 9th of August.
> >
> > Thank you so much for your time.

---

> ### Author Response · Authors · 2022-08-09
> **Reminder for the upcoming deadline**
>
> Dear Reviewer iDC6,
>
> We thank you for your valuable comments. The deadline for the discussion phase is coming soon (within 24 hours). We would like to hear your opinions. If you have any unaddressed concerns, please do not hesitate to discuss them with us.
>
> We appreciate your time and look forward to hearing your ideas.

---

### Author Response · Authors · 2022-08-03
**First Revision**

We thank each of the reviewers for their time and suggestions.

We have uploaded our first revised version of our paper which addresses two important concerns.

In Section 5, we add Remark 5.1 to characterize the local convergence property of Lv.k GP. We also provide a set of experiments on a quadratic game to illustrate the correlations between SPPM and Lv.k GP.

In Section 6, we provide a set of new experiments to demonstrate the efficiency of Lv.k Adam. In addition, we compare it with baseline optimizers in terms of performance (final FID score) and computational cost (average time per iteration).

---

### Meta-Review · Area_Chair_DWKd · 2022-08-25

**Recommendation:** Accept
**Confidence:** Certain

**Metareview:**

This paper proposes a novel recursive reasoning algorithm for minimax games, in which players try to anticipate their opponent's next round move instead of reacting to the current round. Importantly, this is achieved without requiring expensive second order information. Reviewers found the paper clearly written and well motivated, addressing an important problem. The work appears novel, and there is good experimental evidence that the algorithm delivers on its promises.

**Award:**

No

---

### Decision · Program_Chairs · 2022-09-14

Accept